# Sphingosine simultaneously inhibits nuclear import and activates PP2A by binding importins and PPP2R1A

Vaishali Jayashankar[1,8], Peter Kubiniok[2,3,8], Alison N McCracken[1,8], Rebeca G Gentry[1], Kazumi H Eckenstein[1], Lorenzo Sernissi [2], Vito Vece[2], Jean-Baptiste Garsi[2], Sarah Y Valles [1], Sunhee Jung [4], Natalie C Hoffman[1], Arielle S Perrochon[1], Elizabeth M Selwan[1], Brendan T Finicle[1], Mary Pitman[5], DaWei Lin[4], Éric Bonneil[3], Ruijuan Xu[6], Cungui Mao[6], Peter Kaiser [4], David A Fruman[7], David Mobley[5], Cholsoon Jang[4], Stephen Hanessian [2,5 ✉], Pierre Thibault [2,3 ✉] & Aimee L Edinger [1,5 ✉]

## Abstract

**Sphingosine and constrained analogs like FTY720 and SH-BC-893 restrain tumor growth through incompletely defined mechanisms that include protein phosphatase 2A (PP2A) activation. Here we show that these compounds directly bind not only the PP2A scaffolding subunit PPP2R1A, but also the structurally related karyopherins importin-β1 (KPNB1), transportin-1 (TNPO1), importin-5 (IPO5), and importin-7 (IPO7). Binding to sphingosine-like molecules triggers reversible unfolding of these target proteins, resulting in activation of PP2A and inhibition of importins. Although sphingosine engages these proteins, ceramide does not, suggesting that these two endogenous tumor-suppressive sphingolipids work through distinct mechanisms. Simultaneous PP2A activation and importin inhibition reduces nuclear levels of proteins that drive cancer progression and therapeutic resistance such as JUN, YAP, MYC, androgen receptor, hnRNPA1, and NF-κB under conditions where compounds that target PP2A or KPNB1 individually are inactive. These findings provide new insights into sphingolipid biology and highlight a possible path toward cancer therapeutics that could overcome drug resistance.**

**Keywords** Sphingosine; Protein Phosphatase 2A; Nuclear Import; Homeostatic Growth Control
**Subject Categories** Cancer; Membranes & Trafficking; Signal Transduction

## Introduction

Sphingolipids enforce homeostatic growth (Chung et al, 2001; Hannun and Obeid, 2018). Phytosphingosine triggers adaptive quiescence in heat-stressed yeast, while sphingosine and ceramide limit cancer initiation and progression in mammalian cells. Synthetic, sphingosine-like compounds can phenocopy the growth-suppressive effects of natural sphingolipids in both yeast and cancer models. The sphingosine analog FTY720 is an FDA-approved, orally bioavailable immunosuppressant that acts by engaging sphingosine-1-phosphate (S1P) receptors after it is phosphorylated in vivo. However, the unphosphorylated form of FTY720 and phytosphingosine produce largely overlapping transcriptional responses in yeast, and mutants resistant to phytosphingosine are cross-resistant to FTY720 (Welsch et al, 2004). At higher doses than are required for immunosuppression, FTY720 also exhibits potent anti-neoplastic activity in a broad array of preclinical cancer models (Patmanathan et al, 2015). Although FTY720 cannot be repurposed for cancer treatment because FTY720 phosphate triggers profound S1P receptor-dependent bradycardia and immunosuppression (Brinkmann et al, 2010), its anti-cancer effects do not depend on S1P receptors. We generated the cyclic, constrained FTY720 analog SH-BC-893 to avoid S1P receptor engagement, bradycardia, and lymphocyte sequestration while retaining FTY720's anti-neoplastic effects (Chen et al, 2016). SH-BC-893 reduces autochthonous tumor growth in the aggressive *Pb4-Cre;Pten^flox/flox; p53^flox/flox* (prostate double knockout or pDKO) mouse model for castration-resistant prostate cancer by 82% without toxicity to rapidly proliferating normal cells in the bone marrow and intestinal crypts (Kim et al, 2016). In sum, synthetic compounds that mimic the effects of sphingosine and

[1]Department of Developmental and Cell Biology, University of California Irvine, Irvine, CA 92697, USA. [2]Department of Chemistry, Université de Montréal, Montreal, QC H3C 3J7, Canada. [3]Institute for Research in Immunology and Cancer, Université de Montréal, Montreal, QC H3C 3J7, Canada. [4]Department of Biological Chemistry, University of California Irvine School of Medicine, Irvine, CA 92697, USA. [5]Department of Pharmaceutical Sciences, University of California Irvine School of Pharmacy and Pharmaceutical Sciences, Irvine, CA 92697, USA. [6]Department of Medicine, Renaissance School of Medicine, The State University of New York, Stony Brook, NY 11794, USA. [7]Department of Molecular Biology and Biochemistry, University of California Irvine, Irvine, CA 92697, USA. [8]These authors contributed equally: Vaishali Jayashankar, Peter Kubiniok, Alison N McCracken. ✉E-mail: stephen.hanessian@umontreal.ca; pierre.thibault@umontreal.ca; aedinger@uci.edu

phytosphingosine exhibit anti-neoplastic activity and could have potential as cancer therapeutics.

The mechanisms by which sphingosine and these sphingosine-like compounds limit tumor growth are not well defined but likely involve PP2A activation (Patmanathan et al, 2015; Hannun and Obeid, 2018; Kim et al, 2016). FTY720 and the sphingosine derivative ceramide have been reported to activate PP2A by binding to the PP2A inhibitory protein SET (I₂PP2A) (Saddoughi et al, 2013; Mukhopadhyay et al, 2009). However, the FTY720 analog SH-BC-893 and ceramide have distinct effects on the phosphoproteome (Kubiniok et al, 2019) suggesting that ceramide and sphingosine-like compounds may activate PP2A through distinct mechanisms. Moreover, like many natural compounds, sphingolipids may engage multiple targets. Accurately defining the mode of action for anti-neoplastic sphingosine analogs like FTY720 and SH-BC-893 would provide valuable biological insights into the mechanisms underlying homeostatic growth control and highlight effective strategies for cancer therapy. We therefore developed a rigorous orthogonal chemoproteomics strategy using a panel of sphingosine-like compounds and their inactive congeners to identify the shared protein targets of this molecular class.

## Results

### Anti-neoplastic sphingosine-like compounds bind PPP2R1A and a subset of importins

Four anti-proliferative sphingosine-like compounds, SH-BC-893 (Chen et al, 2016), SH-LS-200 (Garsi et al, 2018), FTY720 (Romero Rosales et al, 2011; Kim et al, 2016; Patmanathan et al, 2015), and phytosphingosine (Park et al, 2003; Kang et al, 2017) were modified with an alkyne group at a position that prior structure-activity relationship (SAR) studies suggested would not compromise activity (Garsi et al, 2018; Chen et al, 2016; Garsi et al, 2019; Fransson et al, 2013; Perryman et al, 2016) (Fig. 1A; Appendix Fig. S1A–E). As predicted, each of these affinity ligands was as active in intact cells as the original compound confirming that the alkyne-modified versions retain the ability to bind key targets. To identify and exclude proteins that bind these ligands non-specifically, 6 congeners that were ≥10-fold less active were used as negative controls (Appendix Fig. S1B–F). Compound activity was reduced by N-acetylation (893-NAc, FTY720-NAc, PHS-NAc), eliminating the positive charge of the pyrrolidine nitrogen as a lactam (893-lactam), adding a polar group at the distal end of the hydrocarbon tail (893-ketone), or by introducing polarity in the side chain (SH-LS-200-ketone).

These active and inactive sphingosine-like compounds were immobilized on azide-functionalized beads via click chemistry and incubated with SILAC-labeled cell lysates to allow quantitative comparisons of protein binding (Fig. 1B). After washing, beads coupled to active and inactive congener pairs were combined, bound proteins digested with trypsin, and peptides identified by LC-MS/MS. An enrichment score was calculated for each protein detected in all three biological replicates by expressing binding to active compounds relative to binding to the matched, inactive congener. To identify cell type-specific binding partners, experiments were performed using lysates prepared from two different cell lines, the murine FL5.12 hematopoietic cells used previously for

SAR studies (Garsi et al, 2018; Chen et al, 2016; Garsi et al, 2019; Fransson et al, 2013; Perryman et al, 2016) and the murine prostate cancer epithelial (mPCE) cell line isolated from a prostate tumor in the SH-BC-893-sensitive PTEN/p53 pDKO mouse model (Kim et al, 2016). Proteins that bind selectively to multiple active ligands in both cell lines are likely to be responsible for the anti-neoplastic activity of sphingosine and these sphingosine-like molecules, while proteins that bind equally well or better to the inactive congeners are unlikely to represent biologically relevant targets.

Proteins enriched ≥8-fold by at least one active ligand in at least one cell type are represented in the heat map in Fig. 1C. Among more than 4000 proteins identified in the proteome, five proteins stand out as shared targets of the sphingosine-like molecules examined here: the A scaffolding subunit of PP2A (PPP2R1A) and four karyopherin family members, importin-β1 (KPNB1), transportin-1 (TNPO1), importin-5 (IPO5), and importin-7 (IPO7) (Fig. 1C; Dataset EV1). Although PPP2R1A and these importins lack primary sequence homology, significant structural similarity may explain their binding to the same ligands (Fig. 1D). Intriguingly, when the structure of KPNB1 was first reported in 1999, PPP2R1A was identified as the most structurally similar protein in the PDB database (Vetter et al, 1999). All five of these putative target proteins are scaffolds composed of HEAT (Huntington, elongation factor 3 (EF3), PP2A, and the TOR kinase) repeats. Other karyopherins and HEAT repeat-containing proteins were detected in the FL5.12 and mPCE proteomes but not selectively enriched by this panel of sphingosine-like compounds (Dataset EV1; Appendix Fig. S1G). Interestingly, the four importins that bind these sphingosine-like compounds are functionally related; each can transport ribosomal proteins or the transcription factor JUN into the nucleus (Jäkel and Görlich, 1998; Waldmann et al, 2007). Thus, orthogonal chemoproteomics suggested that PPP2R1A and four importins are the targets of these anti-neoplastic natural and synthetic sphingosine-like compounds.

None of these candidate proteins have previously been identified as a target of sphingosine-like compounds; their isolation here likely reflects improvements in assay design. Prior efforts used affinity ligands modified at sites our SAR studies predict will destroy activity (Haberkant et al, 2016; Saddoughi et al, 2013; Habrukowich et al, 2010). Indeed, the FTY720-biotin used as an affinity ligand in previous studies (Saddoughi et al, 2013) was tenfold less cytotoxic than FTY720 in prostate cancer cells (Appendix Fig. S1H). Prior efforts may also have been compromised by the limited use of negative control compounds and/or biased rather than unbiased protein identification (Haberkant et al, 2016; Habrukowich et al, 2010; Saddoughi et al, 2013). Notably, the PP2A inhibitors SET and ANP32A, reported targets of FTY720 and sphingosine, respectively (Saddoughi et al, 2013; Habrukowich et al, 2010), did not selectively bind any of the four active ligands used in the current study. SET bound equally well to FTY720 and the tenfold less active, acetylated analog FTY720-NAc (Dataset EV1; Appendix Fig. S1C,F,I). Moreover, SET bound strongly to the inactive analog, SH-LS-200-ketone, and not to the active version, SH-LS-200, or to SH-BC-893 (Dataset EV1; Appendix Fig. S1D,F,I). ANP32A was not enriched by any ligand (Dataset EV1). Our data is consistent with a recent report that sensitivity to FTY720 is not correlated with SET expression level across cancer cell lines and that FTY720 remains effective against pancreatic cancer cells after SET is knocked out (Xu et al, 2024). In sum, this unbiased

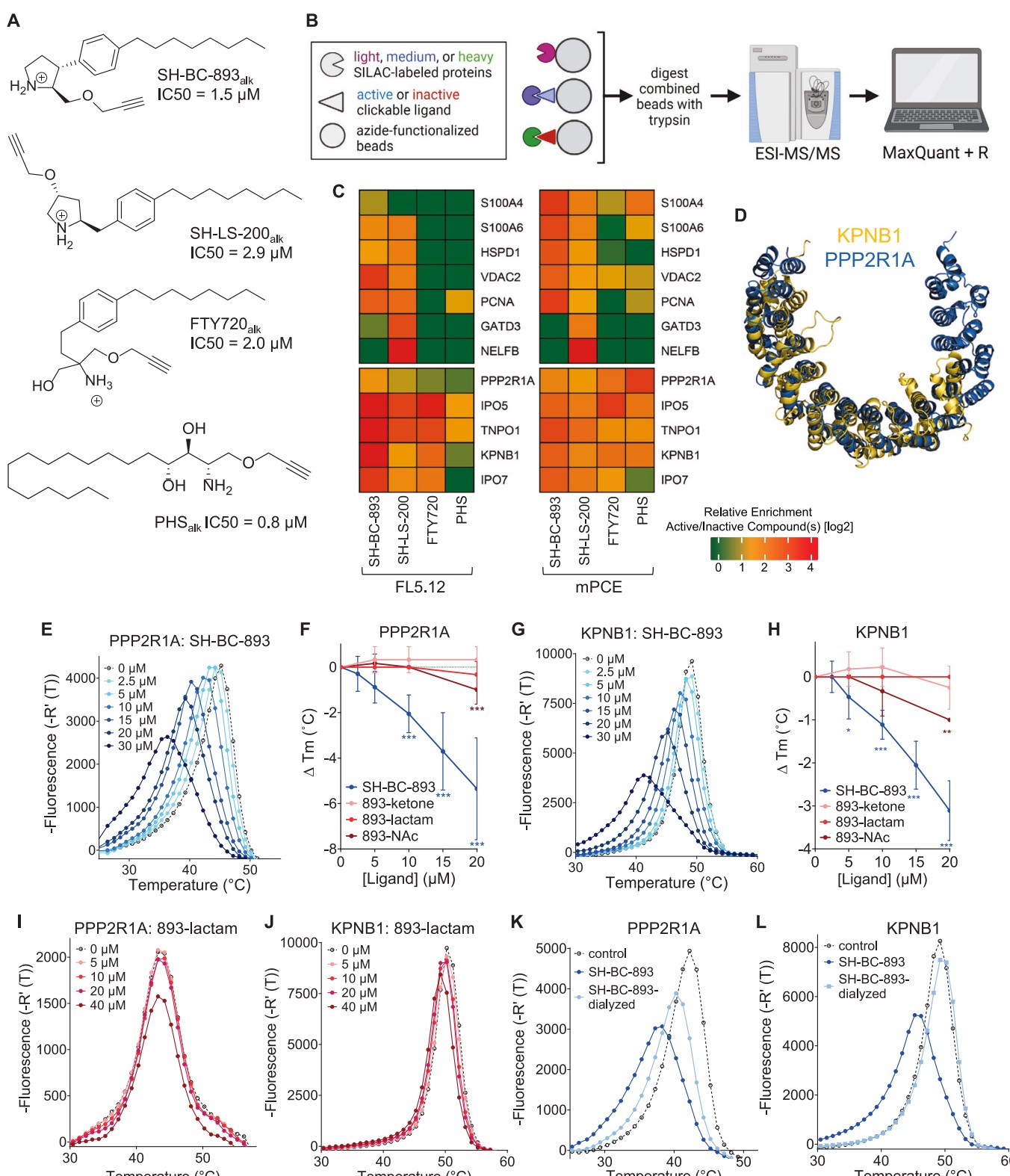

chemoproteomics screen employing multiple negative controls in two distinct cell types suggests that binding to PPP2R1A and/or importins is more likely to explain the anti-neoplastic effects of FTY720 and other sphingosine-like compounds than binding to

SET or ANP32A as previously reported (Saddoughi et al, 2013; Habrukowich et al, 2010).

Subcellular compartmentalization may limit or favor certain ligand-protein interactions in intact cells. To determine whether

**Figure 1.  Sphingosine-like compounds directly bind PPP2R1A and a subset of importins.**

(A) Active ligands shown with their IC50 in FL5.12 cells at 48 h. PHS phytosphingosine, alk alkyne. See also Appendix Fig. S1A–F. (B) Orthogonal chemoproteomics strategy. (C) Proteins enriched by ≥eightfold with a P value ≤ 0.01 by at least one active ligand in FL5.12 and/or mPCE cell lysates. See also Dataset EV1. (D) Overlay of human PPP2R1A (blue, PDB 1B3U) and amino acids 1–442 of human KPNB1 (KPNB1.1; yellow, PDB 1F59). (E–J) Thermal shift assays with recombinant PPP2R1A or KPNB1.1 and the indicated concentrations of active (blue) or inactive (red) ligand. Representative first derivative plots shown in (E, G, I, J). In (F, H), mean ± SD shown; $n = 3$–17, where each biological replicate is a mean of 3–4 technical replicates. Using a one-way ANOVA with Dunnett's multiple comparisons test to compare treated and untreated control samples, ***$P ≤ 0.001$; **$P ≤ 0.01$; *$P ≤ 0.05$; unmarked points not significantly different from control, $P > 0.05$. See Appendix for exact P values. (K, L) Thermal shift assays performed with PPP2R1A (K) or KPNB1.1 (L) and vehicle (dashed line) or SH-BC-893 (dark blue) and repeated after dialysis to remove SH-BC-893 (light blue); $n = 2$, representative plots shown.

sphingosine-like compounds could interact with PPP2R1A and importins in a cellular context, a photoaffinity labeling (PAL) probe was synthesized. Incorporating a diazirine group at the proximal end of the hydrocarbon chain in SH-BC-893-alkyne (893-PAL) did not significantly reduce activity in intact cells suggesting that this probe remains competent to bind key targets (Appendix Fig. 2A). We treated FL5.12 or mPCE cells with 5 μM 893-PAL and performed UV crosslinking to stabilize in situ ligand-protein interactions (Appendix Fig. S2B). After UV irradiation and crosslinking, cells were lysed, azide-biotin covalently linked to the PAL probe-protein complexes via click chemistry, and complexes isolated using streptavidin beads. 893-PAL isolated 203 and 60 proteins from FL5.12 and mPCE cells, respectively (Appendix Fig. S2C; Dataset EV2). Other than PPP2R1A, no PP2A subunits or regulators were enriched by 893-PAL relative to their representation in the proteome (Dataset EV2). KPNB1, TNPO1, IPO5, and IPO7 were among the 20 proteins most significantly enriched by 893-PAL in both cell types. Photoaffinity labeling may also isolate proteins complexed with the ligand-bound target protein. Several proteins that are rapidly dephosphorylated in the presence of SH-BC-893 (e.g., CFL1, EIF4G1, SLC16A1, PDLIM5, and VIM as shown in (Kubiniok et al, 2019)) were bound to 893-PAL consistent with these proteins being PP2A substrates. Karyopherin-interacting proteins, including the RAN GTPase and several heterogeneous nuclear ribonucleoproteins (hnRNPs) that are cargos for TNPO1, were also isolated by 893-PAL (Dataset EV2) (Çağatay and Chook, 2018; Twyffels et al, 2014). Thus, chemoproteomics with cell lysates and photoaffinity labeling in situ both suggest that PPP2R1A and a subset of importins are shared targets of anti-neoplastic, sphingosine-like compounds (Appendix Fig. S2C).

## Sphingosine-like compounds induce conformational changes in PPP2R1A and KPNB1

PPP2R1A and KPNB1 are particularly interesting candidate targets because PP2A activation and KPNB1 inhibition are validated therapeutic approaches in prostate cancer where both SH-BC-893 and FTY720 have demonstrated activity (Yang et al, 2019; Hu et al, 2015; McClinch et al, 2018; Rodriguez-Bravo et al, 2018; Kim et al, 2016; Lim and Wong, 2018). Thermal shift assays were performed to determine whether sphingosine-like compounds bind these two proteins directly. Changes in protein conformation can be monitored using SYPRO Orange, a fluorescent dye that binds to hydrophobic amino acids and is quenched by water (Niesen et al, 2007). Ligand binding often stabilizes proteins, increasing their half-maximal melting temperature ($T_m$). However, SH-BC-893, FTY720, and the natural sphingolipid phytosphingosine reduced

rather than increased the $T_m$ of both recombinant PPP2R1A and KPNB1 (Fig. 1E–H; Appendix Fig. S3A–D). Conformational changes were induced in both proteins at equivalent concentrations of SH-BC-893 confirming that these targets would be engaged in parallel. The inactive congeners of SH-BC-893 failed to induce these conformational changes, and the $T_m$ of a control helical protein, citrate synthase, was not altered by SH-BC-893 indicating that the effect is specific (Fig. 1F,H–J; Appendix Fig. S3E–I). The reduction in $T_m$ likely reflects controlled unfolding rather than non-specific detergent-like effects as Triton X-100 did not alter the $T_m$ of either protein (Appendix Fig. S3E), and conformational changes in PPP2R1A and KPNB1 induced by sphingosine-like compounds were reversible upon removal of the ligand by dialysis (Fig. 1K,L). Thus, these conformational changes in PPP2R1A and KPNB1 are selectively and reversibly induced by active sphingosine-like compounds.

Partial unfolding of PPP2R1A promotes PP2A phosphatase activity (Tsytlonok et al, 2013) providing a potential mechanism for PP2A activation by sphingosine-like compounds. Indeed, the conformational changes induced here by SH-BC-893 were of similar magnitude to those induced by urea in a prior report (Appendix Fig. S3J) (Tsytlonok et al, 2013). Structurally unrelated compounds that activate PP2A, perphenazine (PPZ) and DT-061 (also known as SMAP) (Gutierrez et al, 2014; McClinch et al, 2018), did not alter the $T_m$ of either PPP2R1A or KPNB1 in thermal shift assays suggesting that sphingosine-like compounds activate PP2A via a different mechanism than these compounds (Appendix Fig. S3K,L). Similar to sphingosine-like compounds (Fig. 1G,H; Appendix Fig. S3B,D), the KPNB1 inhibitor importazole (IPZ) also decreases the $T_m$ of KPNB1 (Soderholm et al, 2011). Circular dichroism (CD) spectroscopy offers an alternate means to monitor protein secondary structure. Co-incubation of either PPP2R1A or KPNB1 with SH-BC-893, but not the tenfold less active analog 893-ketone, reduced α-helical structure as reflected by a reduction in ellipticity at 208 nm (Appendix Fig. S3M,N). Thus, CD confirms that direct binding of active sphingosine-like compounds to these proteins triggers conformational changes. Together, these biophysical studies demonstrate that the sphingosine-like compounds selectively and directly bind to PPP2R1A and KPNB1 at equivalent concentrations, triggering conformational changes that may have functional consequences.

To determine where sphingosine-like compounds bind PPP2R1A and KPNB1, 893-diazirine was UV-crosslinked to recombinant PPP2R1A and KPNB1 and both intact and tandem mass analyses performed (Appendix Fig. S4A,B; Datasets EV3–6). As expected, 893-diazirine crosslinked more efficiently to both proteins than the inactive analog 893-NAc-diazirine, although the

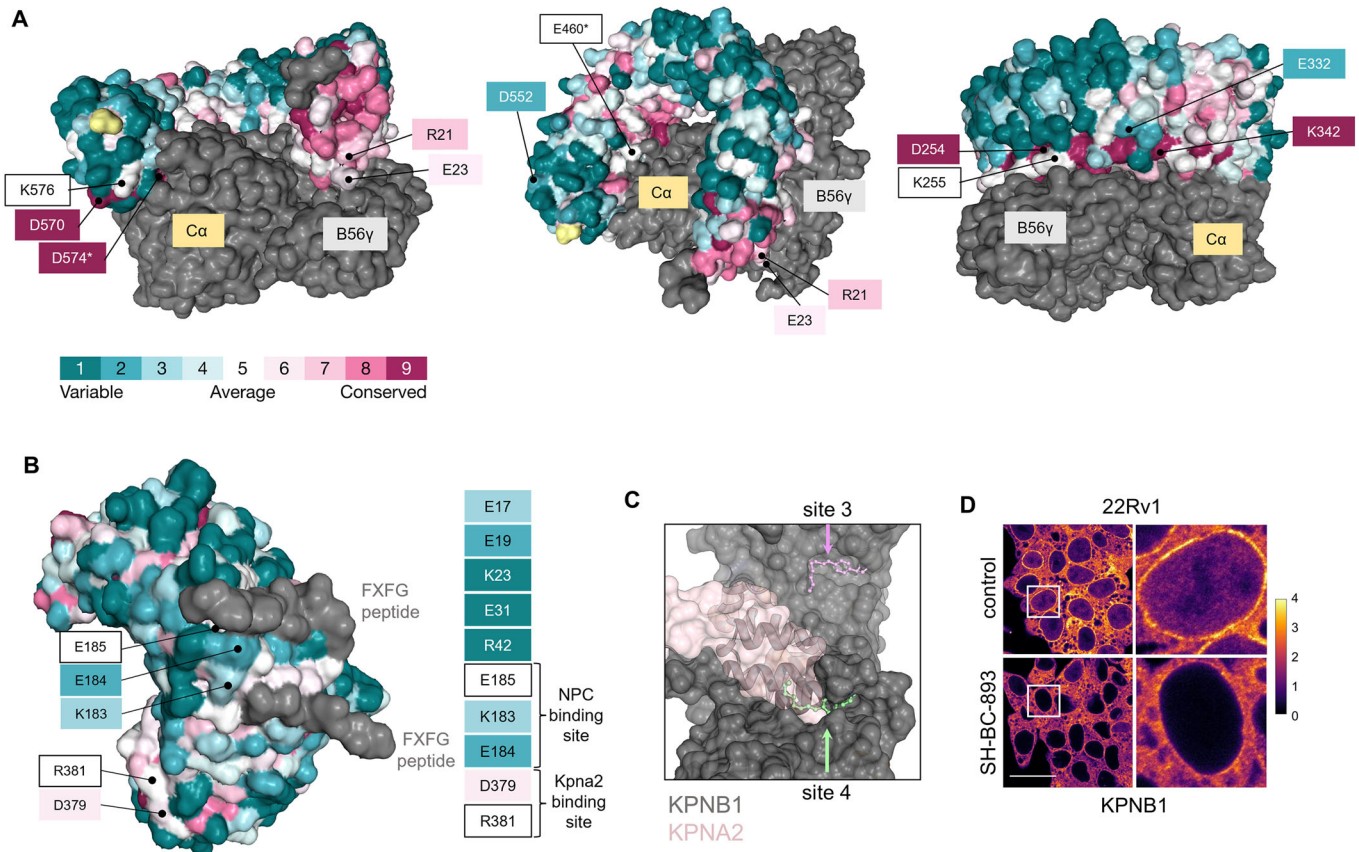

**Figure 2.   893-diazirine crosslinks to PPP2R1A and KPNB1 at functionally important, conserved sites.**

(**A**) Tandem mass spectrometry was used to identify the amino acids in recombinant PPP2R1A that UV-crosslinked to 893-diazirine (see also Appendix Figs. S4B and S5). Conservation determined using ConSurf (Ashkenazy et al, 2016) with default settings, asterisks indicate amino acids that are involved in inter-subunit hydrogen bonding (Cho and Xu, 2007). See also Dataset EV6. Crosslinking sites are shown on PDB 2IAE, a crystal structure of the complex of PPP2R1A (colored to show conservation), PPP2CA (Cα, grey), PPP2R5C (B56γ, grey), and microcystin (not shown). (**B**) As in (**A**) but for KPNB1.1, see also Dataset EV6. PDB 1F59 shown, with FxFG peptides (grey). (**C**) Stable SH-BC-893 binding sites on KPNB1 (PDB 3LWW) predicted by molecular dynamics simulations. KPNA2 (PDB 1QGK, pink). (**D**) Endogenous KPNB1 staining in 22Rv1 cells treated with 10 μM SH-BC-893. Scale bar, 20 μm.

difference was less pronounced for KPNB1 (Appendix Fig. S4B–F; Datasets EV3 and EV4). Unexpectedly, competition experiments using unmodified SH-BC-893, FTY720, or phytosphingosine increased rather than reduced the number of 893-diazirine crosslinks formed (Appendix Fig. S4G,H; Dataset EV5). This result suggests cooperative binding or that initial binding events expose new binding sites. Importantly, the inactive analog 893-lactam did not increase 893-diazirine crosslinking confirming that creating additional binding sites is a specific effect of active ligands.

Tandem mass analyses of tryptic digests identified the cross-linked amino acids (Fig. 2A,B; Appendix Fig. S5; Dataset EV6). Twelve residues in PPP2R1A were crosslinked to 893-diazirine. Each of these amino acids is exposed in the PP2A heterotrimer consistent with activation of the intact phosphatase complex by SH-BC-893 (Fig. 2A) (Cho and Xu, 2007; Finicle et al, 2018; Kubiniok et al, 2019; Kim et al, 2016; Finicle et al, 2023). About half of these crosslinked amino acids are highly conserved and found near the interface with the B or C subunits (Fig. 2A; Appendix Fig. S5) (Ashkenazy et al, 2016; Cho and Xu, 2007). The crosslinked residues E460 and D574 form hydrogen bonds with the C subunit

(Cho and Xu, 2007). Loss of helical content in PPP2R1A, an effect induced by sphingosine-like compounds (Fig. 1E,F; Appendix Fig. S3A,C,J,M), has been reported to activate the complex (Tsytlonok et al, 2013). Increasing the flexibility of the PPP2R1A scaffold could optimize the relative positioning of the B and C subunits to promote catalysis and/or improve B subunit access to substrates in the environment ("fly-casting" model). In sum, further work will be required to determine precisely how sphingosine-like compounds activate PP2A, but PPP2R1A unfolding is a likely mechanism.

893-diazirine crosslinked to ten amino acids in the N-terminal half of KPNB1 that is frequently used in crystallization studies and was utilized here for thermal shift assays (Figs. 1G,H,L and 2B; Dataset EV6). Three of the crosslinked amino acids are conserved. E185 is proximal to the binding site for the FxFG repeats present in a subset of nuclear pore proteins; disrupting contacts between KPNB1 and nuclear pore proteins could prevent translocation through the nuclear pore (Bayliss et al, 2000). However, molecular dynamics simulations performed using the crosslinked amino acids as anchoring points favor a model where SH-BC-893 binds to the

concave surface of KPNB1 near the conserved amino acids D379 and R381 (site 4), a position that might disrupt KPNB1 binding to the importin-α proteins that bridge the interaction with most cargo proteins (Fig. 2B,C; Appendix Fig. S6A,B). Consistent with either of these non-mutually exclusive hypotheses, 893-PAL did not isolate importin-α proteins or nuclear pore proteins (Dataset EV2), and treatment with SH-BC-893 dramatically reduced the amount of KPNB1 present at the nuclear rim and in the nucleus of 22Rv1 prostate cancer cells (Fig. 2D). Together, these data suggest that sphingosine-like compounds inhibit KPNB1 by causing conformational changes that disrupt its interactions with nuclear pore proteins and/or importin-α.

## Anti-neoplastic sphingosine-like compounds reduce nuclear levels of importin cargo proteins

SH-BC-893 is already known to activate PP2A (Kubiniok et al, 2019; Kim et al, 2016; Finicle et al, 2018, 2023). To begin to assess whether SH-BC-893 also inhibits importin function, unbiased nuclear proteomics was performed. High pH reverse phase fractionation of nuclear proteins from SILAC-labeled mPCE cells identified >4,000 non-histone nuclear proteins of which 18% (679 proteins) changed abundance in response to treatment with SH-BC-893 (Fig. 3A and Dataset EV7). Of the nuclear proteins affected by SH-BC-893, 78% were less abundant after treatment suggesting that nuclear import was inhibited. Gene ontology analysis highlighted a depletion of metabolic proteins and proteins involved in cell proliferation and translation from the nucleus (Appendix Fig. S7A). Many ribosomal proteins—which rely on KPNB1, TNPO1, IPO5, and IPO7 for nuclear transport (Jäkel and Görlich, 1998)—are lost from the nuclear proteome and/or co-isolated by the PAL probe consistent with their being cargo for the targeted importins (Table EV1). While the karyopherins responsible for their nuclear import are uncertain, it is striking that the nuclear levels of many metabolic enzymes with "moonlighting" roles in the nucleus (Boukouris et al, 2016; Pan et al, 2021) are reduced in SH-BC-893-treated cells (Appendix Fig. S7A; Table EV2). Although most transcriptional regulators are present in the nucleus at levels below the LC-MS/MS detection limit, unbiased proteomics documented reductions in YAP and JUN (Fig. 3A; Dataset EV7), nuclear proteins linked to prostate cancer progression and recurrence (Nguyen et al, 2015; Ouyang et al, 2008; Riedel et al, 2021; Lee et al, 2021; Thakur et al, 2014; Edwards et al, 2004; Kuser-Abali et al, 2015). The loss of YAP and JUN from the nucleus of SH-BC-893-treated mPCE cells was confirmed by immunofluorescence microscopy (Fig. 3B,C). The nuclear translocation of YAP depends upon IPO7 (García-García et al, 2022) while JUN is carried into the nucleus by the four importins targeted by SH-BC-893: KPNB1, TNPO1, IPO5 and IPO7 (Waldmann et al, 2007) (Appendix Fig. S7B). Although cytosolic JUN was not detectable by microscopy after treatment with SH-BC-893, total cellular JUN levels remained constant as determined by western blotting (Appendix Fig. S7C). Immunofluorescence microscopy confirmed that all four active sphingosine-like compounds used for chemoproteomics reduced nuclear levels of YAP and JUN in mPCE cells while their inactive congeners had either no or greatly diminished effects (Appendix Fig. S7D,E). SH-BC-893 also reduced nuclear YAP levels in MYC-CaP, LNCaP, and 22Rv1 prostate cancer cells (Fig. 3D,E; Appendix Fig. S8A) demonstrating that the effect is not restricted to mPCE cells. These data suggest that SH-BC-893 inhibits importin function.

A caveat to this interpretation is that some oncogenic transcription factors are degraded in response to PP2A activation. To determine whether PP2A activation alone could be sufficient to account for the observed changes in YAP and JUN nuclear localization, we measured the effect of the PP2A activator PPZ on the nuclear levels of YAP and JUN. Although PPZ slightly reduced nuclear YAP levels, its effects were not as profound as those of SH-BC-893 or other sphingosine-like compounds (Fig. 3B,C; Appendix Fig. S7D). PPZ did not reduce nuclear levels of JUN (Fig. 3B,C). Importantly, this concentration of PPZ reduced the nuclear levels of MYC, a protein that is degraded in response to PP2A-dependent dephosphorylation (Gutierrez et al, 2014; Ruvolo, 2016), as efficiently as SH-BC-893 (Fig. 3B,C). Moreover, SH-BC-893 but not PPZ reduced nuclear levels of the TNPO1 cargo and cancer-associated protein hnRNPA1 (Fig. 3B,C; Appendix Fig. S7B) (Möller et al, 2020; Twyffels et al, 2014; Roy et al, 2017). Taken together, these results suggest that SH-BC-893, but not PPZ, inactivates importins.

## Parallel actions on PP2A and KPNB1 make SH-BC-893 a robust inhibitor of MYC and AR

The prostate cancer drivers MYC and the androgen receptor (AR) are degraded in response to PP2A-dependent dephosphorylation, but also rely on KPNB1 to reach their target genes in the nucleus (Antonarakis et al, 2014; Kuser-Abali et al, 2015; Labbé and Brown, 2018; Rodriguez-Bravo et al, 2018; McClinch et al, 2018; Sandhu et al, 2021; Ruvolo, 2016) (Appendix Fig. S7B). As expected of a molecule that both activates PP2A and inhibits KPNB1, SH-BC-893 reduced nuclear levels of MYC in mPCE, MYC-CaP, LNCaP, and 22Rv1 prostate cancer cell lines and autochthonous tumors from pDKO mice lacking PTEN and p53 in the prostate (Fig. 3B–D,F; Appendix Fig. S8A–C). SH-BC-893 also inhibited the androgen-induced nuclear translocation of the AR (Fig. 3D,G; Appendix Fig. S8A, AR expression in mPCE cells was below the limit of detection). The profound decrease in nuclear AR in 22Rv1 cells is noteworthy as this cell line expresses AR-V7, a ligand-independent AR splice variant linked to drug resistance in patients with metastatic castration-resistant prostate cancer (Antonarakis et al, 2014; Sharp et al, 2019). The ability of SH-BC-893 to reduce nuclear levels of AR-V7 was confirmed in HeLa cells expressing GFP-AR-V7 (Appendix Fig. S8D,E). Quantitative RT-PCR indicated that these changes in nuclear transcription factor levels were sufficient to alter target gene expression. SH-BC-893 decreased the level of mRNAs induced by MYC, JUN, or the AR and increased the levels of mRNA repressed by these transcription factors (Appendix Fig. S8F,G). These data confirm that SH-BC-893 limits the activity of MYC and the AR in multiple cell lines as expected. However, they cannot distinguish whether inactivation stems from PP2A activation, KPNB1 inhibition, or both.

Polypharmacology is a validated strategy to overcome drug resistance (Vincent et al, 2022; Knight et al, 2010). Engaging two anti-neoplastic pathways in parallel should make SH-BC-893 effective in cells that have been rendered resistant to compounds that selectively activate PP2A (PPZ) (Gutierrez et al, 2014) or inhibit KPNB1 (importazole, IPZ) (Soderholm et al, 2011; Rodriguez-Bravo et al, 2018) (Fig. 4A). To test this prediction,

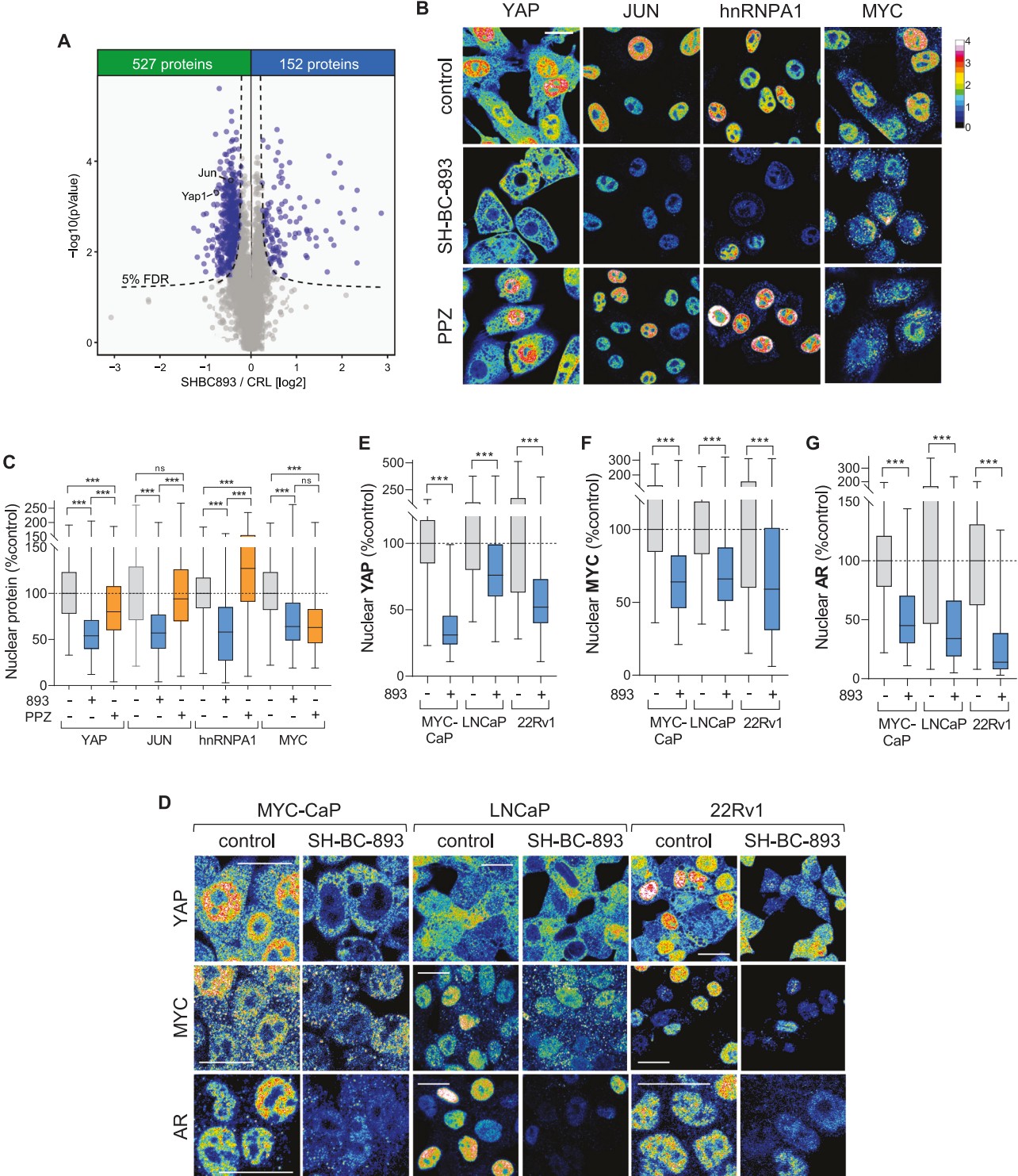

PP2A-dependent proteasomal degradation was blocked with epoxomicin. As expected, PPZ was no longer effective in the presence of epoxomicin (Fig. 4A–D; Appendix Fig. S9A). In contrast, proteasome inhibition did not diminish the ability of SH-BC-893 to reduce nuclear MYC and AR levels consistent with parallel inhibition of KPNB1-dependent nuclear import. To complement these studies with epoxomicin, PP2A was disabled

by acutely eliminating PPP2R1A using the auxin-inducible degron system (Fig. 4E) (Natsume et al, 2016; Li et al, 2019). The scaffolding subunit is essential for PP2A activity, and PPP2R1A is the predominant isoform (Zhou et al, 2003). A FLAG-tagged auxin-inducible degron (AID) was fused to the N-terminus of both alleles of PPP2R1A in 293T cells using CRISPR/Cas9-mediated microhomology-mediated end joining (Appendix Fig. S9B,C). In

**Figure 3. Constrained cyclic sphingosine analog SH-BC-893 reduces nuclear levels of cargo proteins transported by KPNB1, TNPO1, IPO5, and IPO7.**

(A) Non-histone nuclear proteins in mPCE cells whose levels were altered by a 6 h treatment with 5 µM SH-BC-893 as determined by LC-MS/MS. Mean of four biological replicates shown. Significance was calculated using a $t$ test. Volcano plot shows $\log_2$(fold change) vs $-\log_{10}(P$ value). The curved significance threshold represents a false discovery rate (FDR)-adjusted boundary at 5%. See also Dataset EV7. (B, C) Endogenous YAP, JUN, hnRNPA1, or MYC localization in mPCE cells following a 6 h treatment with 5 µM SH-BC-893 or 20 µM perphenazine (PPZ). (D–G) YAP, MYC, or AR localization in the indicated cell lines treated with 15 µM (MYC-CaP and LNCaP) or 10 µM (22Rv1) SH-BC-893. All cells were maintained in 10 nM dihydrotestosterone (DHT) where AR is evaluated, LNCaP cells were also maintained in DHT during YAP and MYC evaluation. In (D), fluorescence intensity scale is as in (B). For (D), larger field of view shown in Appendix Fig. S8A. Box-and-whisker plots display the median and interquartile range (box), as well as the range of the lower and upper quartiles (whiskers). In (C, E–G), $n > 200$ cells from two biological replicates. As a visual inspection and a D'Agostino & Pearson test indicated that the data was not normally distributed, a Kruskal–Wallis with Dunn's multiple comparisons test (C) or Mann–Whitney test (E–G) was applied; ***$P \le 0.001$ or ns, not significant $P > 0.05$. See Appendix for exact $P$ values. Scale bars, 20 µm.

the absence of the auxin 1-naphthaleneacetic acid (NAA), SH-BC-893 and PPZ both reduced nuclear MYC levels in 293T cells (Fig. 4F,G). Following the NAA-induced degradation of PPP2R1A, PPZ was no longer effective (Fig. 4E–G). However, SH-BC-893 reduced nuclear MYC levels to the same extent in the presence or absence of PPP2R1A consistent with parallel inhibition of KPNB1 (Fig. 4A,F,G). Thus, SH-BC-893 continues to block nuclear import in cells rendered resistant to selective PP2A activation.

Next, we evaluated whether KPNB1 over-expression rendered cells resistant to KPNB1 inhibitors. Because constitutive KPNB1 over-expression was toxic to cells, a doxycycline-regulated transgene was used (Fig. 4H,I). As expected, KPNB1 over-expression prevented the reduction in nuclear MYC levels normally produced by the selective KPNB1 inhibitor IPZ (Fig. 4I,J). IPZ blocks nuclear import but does not trigger MYC degradation (Fig. 4A), and MYC accumulated in a peri-nuclear compartment when it was excluded from the nucleus upon KPNB1 inhibition with IPZ (Fig. 4I). In contrast, SH-BC-893 reduced nuclear MYC levels to an equal extent in control and KPNB1 over-expressing cells (Fig. 4I,J) consistent with parallel PP2A activation (Fig. 4A). When KPNB1 was over-expressed and PP2A-dependent proteasomal degradation was blocked with epoxomicin, SH-BC-893 no longer reduced nuclear MYC (Appendix Fig. S9D,E). These experiments demonstrate that SH-BC-893 reduces nuclear levels of prostate cancer drivers MYC and the AR by triggering PP2A-dependent proteasomal degradation and blocking KPNB1-mediated nuclear import in parallel, redundancy that could limit the development of drug resistance.

## Sphingosine, but not ceramide, engages PPP2R1A and importins

We hypothesized that SH-BC-893 and the other sphingosine-like compounds we evaluated (Fig. 1A) engage the same protein targets as endogenous sphingosine. Indeed, sphingosine triggered conformational changes in both PPP2R1A and KPNB1 in the same concentration range as SH-BC-893 and FTY720 (Figs. 1E–H and 5A,B; Appendix Fig. S3A,B); notably, the enzymes that metabolize sphingosine are not present in these thermal shift assays using recombinant proteins. Moreover, C2-, C6-, and C16-ceramide did not trigger conformational changes in PPP2R1A or KPNB1 even at high concentrations (Fig. 5A,B; Appendix Fig. S9F–H). The inactivity of N-acylated forms of sphingosine (ceramides) is consistent with the observation that N-acetylated sphingosine-like compounds did not bind to PPP2R1A or importins in chemoproteomics assays (Fig. 1C; Dataset EV1; Appendix Fig. S1G) and failed to alter the $T_m$ of PPP2R1A and KPNB1 in thermal shift

assays (Fig. 1F,H; Appendix Fig. S3G,I). To confirm that ceramide does not inhibit importin function, nuclear MYC levels were evaluated in 22Rv1 cells where PP2A-dependent MYC degradation was blocked with epoxomicin. As predicted, PP2A activation with PPZ or ceramide reduced nuclear MYC in a proteasome-dependent manner (Appendix Fig. S9I,J). In contrast, SH-BC-893 remained effective in the presence of epoxomicin consistent with its ability to activate PP2A and inhibit KPNB1 in parallel (Fig. 4A; Appendix Fig. S9I,J). These results are also consistent with the differential effects of ceramide and SH-BC-893 on the phosphoproteome (Kubiniok et al, 2019). Ceramide treatment results in much greater dephosphorylation of nuclear proteins than does exposure to SH-BC-893 which also inhibits nuclear import. In sum, these results suggest that while sphingosine and ceramide both suppress tumor growth (Hannun and Obeid, 2018; Guenther et al, 2008; Kubiniok et al, 2019), they likely do so via distinct mechanisms.

To assess whether sphingosine reduced importin activity in intact cells, sphingosine metabolism was disrupted. Exogenous sphingosine is rapidly metabolized to S1P but can be stabilized by sphingosine kinase 1 (SK1) deletion (Fig. 5C–F). It was necessary to add exogenous sphingosine to SK1 knockout (KO) murine embryonic fibroblasts (MEFs) because cells compensate for chronic SK1 inactivation; basal sphingosine levels are not elevated in SK1 KO MEFs (Appendix Fig. S10A). Although exogenous sphingosine had little effect in SK1 wild-type (WT) MEFs, nuclear levels of the IPO7 cargo YAP were significantly reduced in SK1 KO MEFs where sphingosine was stable over the 6 h of treatment (Fig. 5D–H). Sphingosine is known to activate PP2A. Because PP2A activation with PPZ reduced nuclear YAP levels in SK1 KO MEFs to a similar extent as sphingosine (Fig. 5I,J), PP2A-dependent proteasomal degradation was blocked with epoxomicin. As expected, PP2A activation with PPZ no longer reduced nuclear YAP when proteasomal degradation was blocked. However, both SH-BC-893 and sphingosine reduced nuclear YAP levels even in epoxomicin-treated SK1 KO cells consistent with parallel IPO7 inhibition. A selectively KPNB1-dependent cargo was next evaluated (Appendix Fig. S7B). Although MYC was expressed at a level below the detection limit in SK1 KO MEFs, the TNFα-induced nuclear translocation of the KPNB1-dependent cargo NF-κB (Liang et al, 2013) was readily detectable (Fig. 5K–M). As expected, SH-BC-893 completely blocked NF-κB nuclear translocation in response to TNFα in both WT and SK1 KO MEFs to a similar extent. Consistent with its enhanced metabolic stability (Fig. 5E), sphingosine blocked the nuclear translocation of NF-κB more efficiently in SK1 KO than SK1 WT MEFs (Fig. 5K–M). Because NF-κB nuclear translocation can be detected after just 30 min, sphingosine was more effective in SK1 WT MEFs in this assay than in

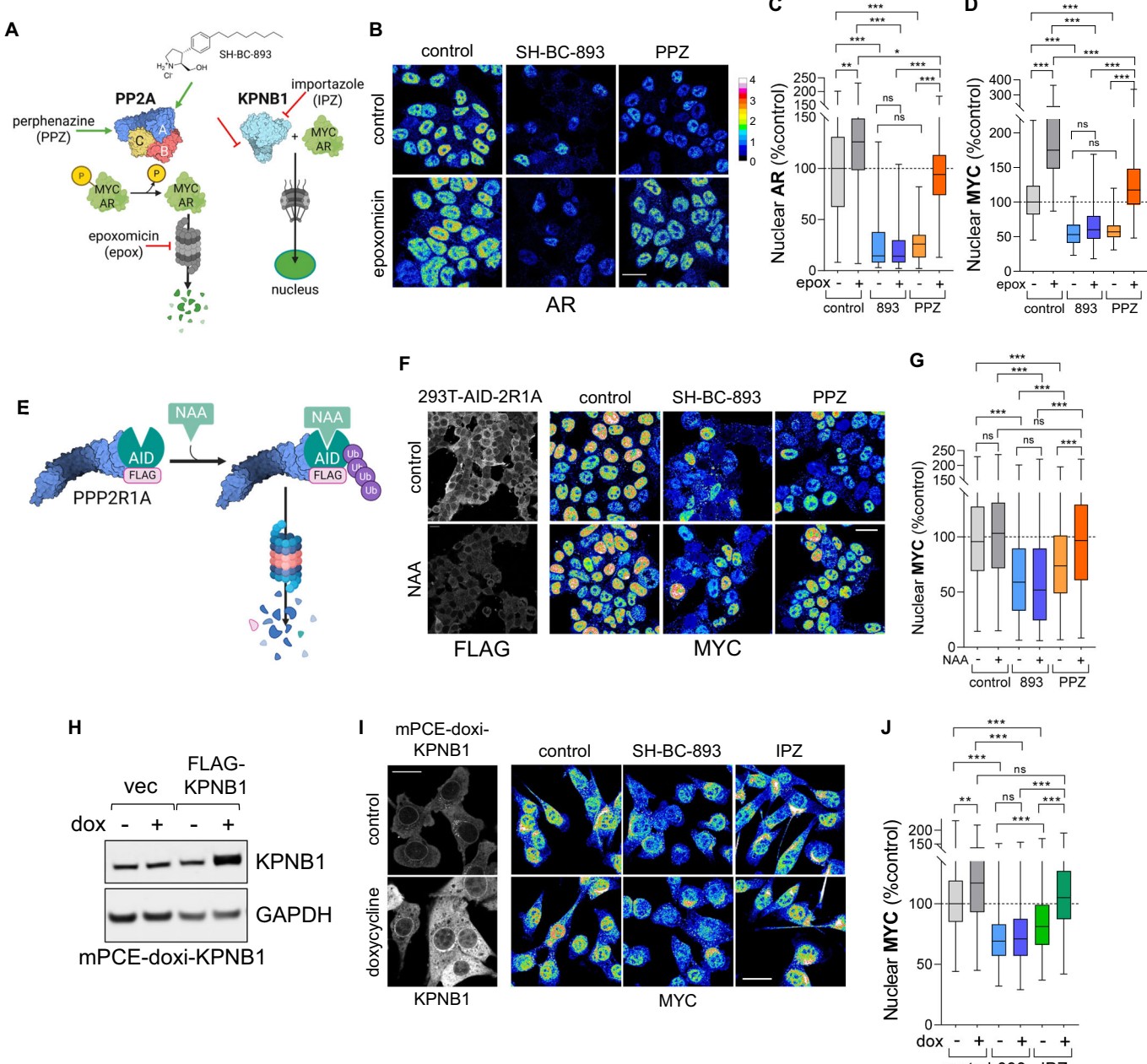

**Figure 4. The parallel actions of SH-BC-893 on PPP2R1A and KPNB1 contribute to MYC and AR inhibition.**

(A) Model for SH-BC-893 inhibition of MYC and AR through redundant pathways. PPZ, perphenazine; epox, epoxomicin; IPZ, importazole. Endogenous AR (B, C) or MYC (D) and Appendix Fig. S9A) localization in 22Rv1 cells treated with 10 μM SH-BC-893 or 25 μM PPZ ± 100 nM epoxomicin for 6 h. In (C, D), n > 140 cells from two biological replicates. (E) Model showing mechanism for FLAG-PPP2R1A degradation in response to the auxin NAA in homozygous knock-in cells. See also Appendix Fig. S9B,C. (F, G) FLAG-PPP2R1A or endogenous MYC localization in 293T-AID-2R1A cells treated with SH-BC-893 (10 μM) or PPZ (25 μM) for 6 h ± a 3 h pre-treatment with 500 μM NAA. In (G), n > 150 cells from two biological replicates. (H) Western blot validating doxycycline-inducible over-expression of KPNB1 in mPCE cells. Representative blot shown, n = 3. (I, J) KPNB1 or endogenous MYC localization in mPCE-doxi-KPNB1 cells treated for 6 h with SH-BC-893 (5 μM) or IPZ (40 μM) ± an overnight treatment with 1 μg/ml doxycycline. In (J), n > 200 cells from two biological replicates. Box-and-whisker plots display the median and interquartile range (box), as well as the range of the lower and upper quartiles (whiskers). In (F, I), fluorescence intensity scale for nuclear proteins as in (B). In (C, D, G, J) a visual inspection and a D'Agostino & Pearson test indicated that the data was not normally distributed and a Kruskal–Wallis test with Dunn's multiple comparisons test was applied; ***P ≤ 0.001; **P ≤ 0.01; *P ≤ 0.05; ns, not significant P > 0.05. See Appendix for exact P values. Scale bars, 20 μm.

(Fig. 5G,H) where 6 h are required to allow for YAP export from the nucleus after YAP import is interrupted due to IPO7 inhibition. Supporting the proposal that the interaction between sphingosine and these targets is specific, the precursor 3-keto-sphinganine did not efficiently trigger conformational changes in PPP2R1A or KPNB1, and shortening, lengthening, or altering the position of the double bond in the hydrocarbon chain of sphingosine altered target engagement (Appendix Fig. S10B,C). Together, these results

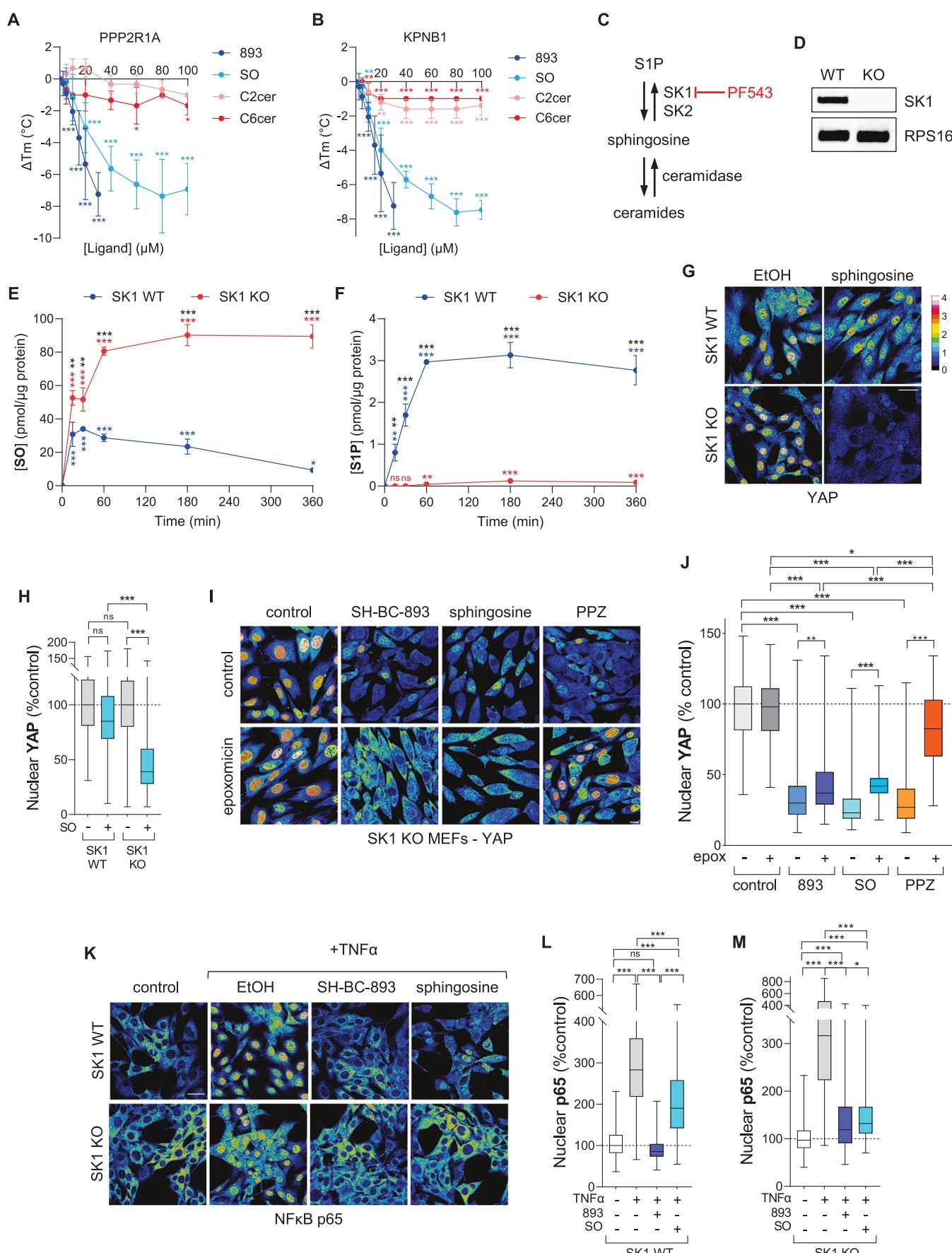

◀

**Figure 5. Sphingosine, but not ceramide, binds PPP2R1A and inhibits importins.**

(A, B) Recombinant PPP2R1A (A) or KPNB1.1 (B) binding to SH-BC-893 (893), sphingosine (SO), C2-ceramide (C2cer), or C6-ceramide (C6cer) was evaluated in thermal shift assays; mean ± SD shown, $n = 3$–17 where each biological replicate is a mean of 3–4 technical replicates. (C) Metabolic pathways regulating sphingosine levels. SK1/2, sphingosine kinase 1 or 2. (D) Validation of SK1 WT and KO MEFs by RT-PCR, representative gel shown, $n = 2$. (E, F) d18:1 sphingosine (E) or S1P (F) levels over time in wild-type (WT) or SK1 KO MEFs treated with 10 μM d18:1 sphingosine as measured by LC-MS/MS; mean ± SD, $n = 3$. (G, H) YAP localization in SK1 WT or KO MEFs treated for 6 h with 10 μM sphingosine. In (H), $n > 100$ cells from two biological replicates. (I, J) YAP localization in SK1 KO MEFs treated with SH-BC-893 (15 μM), sphingosine (15 μM), or PPZ (25 μM) ± epoxomicin (100 nM) for 3 h. In (J), $n > 200$ cells from two biological replicates. (K–M) NF-κB (p65) localization in SK1 WT or KO MEFs treated with 20 nM TNF-α and vehicle (ethanol), SH-BC-893 (5 μM), or sphingosine (10 μM) for 1 h. In (L, M), $n > 170$ cells from two biological replicates. Box-and-whisker plots display the median and interquartile range (box), as well as the range of the lower and upper quartiles (whiskers). In (I, K), intensity scale same as in (G). Using a one-way ANOVA with Dunnett's multiple comparisons test to compare to the untreated control (A, B), or a D'Agostino and Pearson normality test followed by a Kruskal–Wallis test with Dunn's multiple comparisons test (H, J, L, M), ***$P \leq 0.001$; **$P \leq 0.01$; *$P \leq 0.05$; ns, not significant $P > 0.05$. See Appendix for exact $P$ values. In (A, B), asterisks are colored to match the compound being compared to its untreated control. In (E, F), black asterisks compare WT and KO with multiple $t$ tests, red or blue asterisks compare individual time points for each cell line with its own untreated control using an ANOVA. Scale bars, 20 μm.

indicate that, like SH-BC-893, sphingosine inhibits IPO7- and KPNB1-dependent nuclear import.

Although exogenous sphingosine inhibited importin function (Fig. 5G–M), it was unclear whether endogenous sphingosine levels could rise to sufficient levels to engage these target proteins. To increase endogenous sphingosine levels, SK1 was acutely inhibited with a small molecule, PF-543 (Schnute et al, 2012) (Figs. 5C and 6A). Because many inhibitors of sphingolipid-metabolizing enzymes are sphingosine analogs that might directly engage the same targets as sphingosine, we first confirmed that PF-543 did not engage PPP2R1A or KPNB1. While the first-generation SK1 inhibitors safingol and dimethylsphingosine (DMS) induced conformational changes in PPP2R1A and KPNB1 at the same concentrations that they are used to inhibit SK1 (Appendix Fig. S10D–H) (Edsall et al, 1998; Dickson et al, 2011; Olivera et al, 1998), PF-543 did not alter the $T_m$ of either PPP2R1A or KPNB1 even at 100 μM (Fig. 6A,B). Thus, PF-543 can be used to alter endogenous sphingosine levels without directly affecting PPP2R1A or KPNB1.

LC-MS/MS confirmed that acute SK1 inhibition with PF-543 elevated endogenous sphingosine levels. As expected, PF-543 raised endogenous sphingosine levels threefold and reduced S1P levels in HeLa cervical cancer cells (Fig. 6C,D). Consistent with this, PF-543 reduced TNFα-induced nuclear translocation of NF-κB (Fig. 6E,F). Based on the volume (median = 2327 fL) and average protein content (333 pg/cell) of our HeLa cells, treatment with PF-543 raised the intracellular sphingosine concentration from 16 μM to 47 μM (Appendix Fig. S10A). This value is consistent with cellular sphingosine levels reported in the literature (Schnute et al, 2012; Merrill, 1991; Chao et al, 1992; Becker et al, 2005; Bai et al, 2019; Taha et al, 2004; Xu et al, 2016; Lima et al, 2017; Merrill et al, 1988; Zitomer et al, 2009) and our measurements in other cells used in these studies (Appendix Fig. S10A) demonstrating that the concentration of exogenous sphingosine and sphingosine-like compounds used in these studies (2–15 μM) is within the physiologically relevant range. The PF-543-induced changes in sphingosine levels observed here are also of similar magnitude to the increases in sphingosine reported in SK1 KO mice that have reduced tumor burdens (Kohno et al, 2006; Chen et al, 2018b).

To confirm these results with the small molecule PF-543, endogenous sphingosine levels were also elevated using a genetic method. To overcome the compensation that can occur with chronic over-expression of sphingosine-metabolizing enzymes, a doxycycline-inducible expression system was used to over-express alkaline ceramidase 1 (ACER), an enzyme that produces

sphingosine from ceramide (Fig. 5C) (Sun et al, 2009). Peak expression of FLAG-tagged ACER1 was obtained within 24 h of induction, a time point at which endogenous sphingosine levels were increased approximately fivefold (Fig. 6G; Appendix Fig. S10A,I). ACER1 was localized to the endoplasmic reticulum as expected (Appendix Fig. S10J). Consistent with results in SK1 KO MEFs treated with exogenous sphingosine (Fig. 5G–M), acute expression of ACER1 reduced nuclear levels of JUN and YAP similar to SH-BC-893 (Fig. 6H–J). Taken together, the results reported here indicate that both sphingosine and sphingosine-like anti-neoplastic compounds activate PP2A through PPP2R1A while directly inhibiting KPNB1, IPO7, TNPO1, and IPO5 in parallel.

## Discussion

Using chemoproteomics (Fig. 1B,C; Dataset EV1), thermal shift assays (Figs. 1E–L and 5A,B; Appendix Fig. S3A–I), and cross-linking studies (Fig. 2A,B; Appendix Figs. S2, S4, and S5; Datasets EV2–EV6), we discovered and validated unexpected targets for sphingosine and sphingosine-like compounds: the structurally homologous proteins PPP2R1A, KPNB1, TNPO1, IPO7, and IPO5. Obtaining consistent results using four structurally related and biologically active affinity ligands in two different cell lines, plus using a photoaffinity labeling probe in intact cells as a complementary approach (Fig. 1A,B; Datasets EV1 and EV2; Appendix Figs. S1A–F and S2), creates confidence that these targets are responsible for the shared anti-neoplastic activity of this molecular class. Although PP2A activation by sphingosine-like compounds is established (Patmanathan et al, 2015; Finicle et al, 2018; Romero Rosales et al, 2011; Kubiniok et al, 2019; Kim et al, 2016; Finicle et al, 2023) the interaction with PPP2R1A is a new finding. Similarly, while the controlled unfolding of PPP2R1A was already shown to activate the heterotrimer using urea (Tsytlonok et al, 2013), establishing that an endogenous ligand can trigger this conformational change (Fig. 5A; Appendix Fig. S3C) now suggests that this unfolding could be a physiologically relevant mode of PP2A regulation. PPP2R1A binding by sphingosine-like compounds might have been anticipated given their activation of PP2A, but the association with importins was completely unexpected. Sphingosine-like compounds had not been previously linked to nuclear transport. Inhibition of nuclear import by sphingosine and related compounds is supported by: the reduced association of KPNB1 with the nuclear pore and nucleus (Fig. 2D), unbiased nuclear proteomics showing depletion of many proteins associated

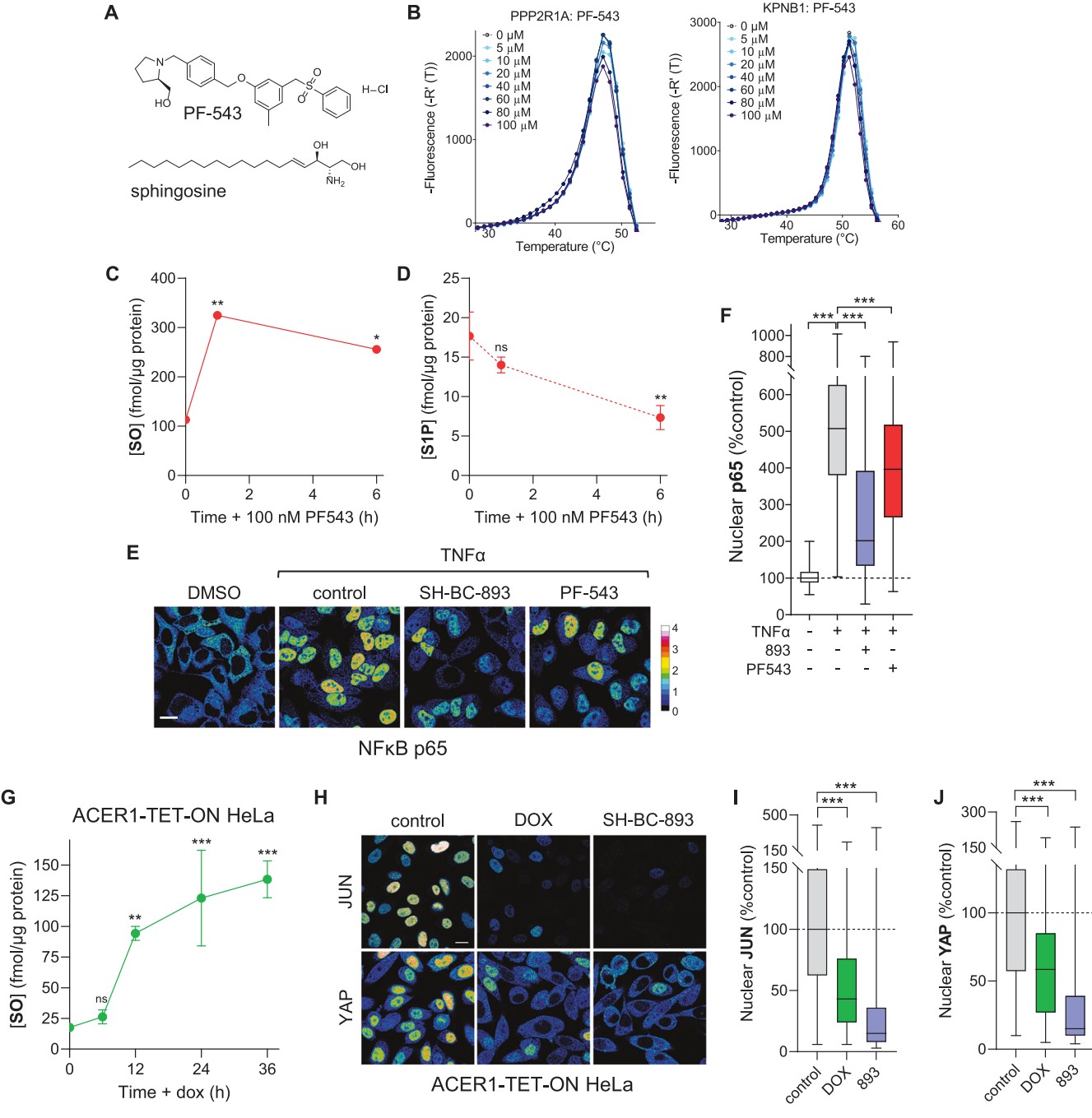

**Figure 6. Endogenous sphingosine inhibits importins.**

(A) Structures of PF-543 and sphingosine. (B) Thermal shift assays using recombinant PPP2R1A or KPNB1.1 and the indicated concentrations PF-543. Representative first derivative plots shown; $n = 3$. (C, D) Endogenous sphingosine (SO) (C) or S1P (D) levels in HeLa cells treated for the indicated time with PF-543 (100 nM) measured by LC-MS/MS; mean ± SD, $n = 3$. (E, F) NF-κB localization in HeLa cells treated with 100 ng/ml TNFα for 30 min. Cells were treated with 100 nM PF-543 for 6 h. In (F), $n > 150$ cells from two biological replicates. (G) ACER1 expression was induced in HeLa cells with 10 ng/ml doxycycline for the indicated time interval and sphingosine levels measured by LC-MS/MS; mean ± SD, $n = 3$. (H–J) JUN and YAP localization in ACER1-TET-ON HeLa cells treated for 6 h with SH-BC-893 (10 μM) or doxycycline (10 ng/ml) for 24 h. In (I, J), $n > 500$ cells from two biological replicates. In (H), fluorescence intensity scale for nuclear proteins as in (E). Box-and-whisker plots display the median and interquartile range (box), as well as the range of the lower and upper quartiles (whiskers). Using a one-way ANOVA with Dunnett's multiple comparisons test (C, D, G), a D'Agostino and Pearson normality test followed by a Kruskal–Wallis test with Dunn's multiple comparisons test (F, I, J), ***$P \leq 0.001$; **$P \leq 0.01$; *$P \leq 0.05$; ns, not significant $P > 0.05$. See Appendix for exact $P$ values. Scale bars, 20 μm.

with oncogenesis from the nucleus (Fig. 3A; Appendix Fig. S7A), reduced nuclear levels of YAP, JUN, hnRNPA1, MYC, and AR even after blocking PP2A-dependent proteasomal degradation (Figs. 3, 4, and 5H,I; Appendix Fig. S9I,J), and inhibition of the ligand-stimulated nuclear translocation of the AR and NF-κB (Figs. 3D, G, 5K–M, and 6E,F). Interestingly, IPZ also reduces the $T_m$ of KPNB1 (Soderholm et al, 2011). While its mode of action is uncertain, the same study showed that IPZ induces a conformational change in the KPNB1-RAN-GTP complex without causing its dissociation. This mechanism of action could also be consistent with our data with SH-BC-893; RAN was isolated by the 893-PAL probe (Dataset EV2) suggesting that SH-BC-893 binds to importins that are associated with RAN. In sum, this study provides new insights into the biological actions of sphingosine and a series of anti-neoplastic constrained sphingosine analogs that should open new research directions.

Parallel PP2A activation and importin inhibition would have profound effects on cell physiology. PP2A not only opposes many oncogenic signaling pathways (Ruvolo, 2016), it also limits nutrient access through its effects on endolysosomal trafficking (Kim et al, 2016). Moreover, these four importins ferry a wide range of cargos into the nucleus thereby supporting pro-growth transcriptional programs (Çağatay and Chook, 2018). Functional overlap may explain why these four importins are affected; KPNB1, TNPO1, IPO5, and IPO7 are responsible for the nuclear import of JUN and ribosomal proteins (Waldmann et al, 2007; Jäkel and Görlich, 1998). These multifaceted effects on PP2A and nuclear import suggest that sphingosine may have a more significant role in sphingolipid-mediated homeostatic growth control than is currently appreciated, particularly given that ceramide does not engage these targets (Fig. 5A,B; Appendix Fig. S9F–J). Ceramide is N-acylsphingosine, and N-acetylation of multiple sphingosine-like compounds also disrupted binding to PPP2R1A and importins (Fig. 1C,F,H; Appendix Figs. S1A–G, S4A–F, and S7D,E). Both sphingosine and ceramide are recognized as tumor-suppressive lipids, but ceramide's actions are much more commonly studied; ceramide and S1P levels are routinely measured, but sphingosine levels are seldom monitored. This focus on ceramide and S1P can be attributed in part to the pervasive sphingolipid rheostat model where S1P promotes and ceramide suppresses growth with sphingosine playing a redundant and subordinate role to ceramide. The present study clarifies that sphingosine and ceramide have distinct molecular modes of action. It also provides a strong rationale for both monitoring sphingosine levels and considering its unique effects on nuclear transport in future experiments where sphingolipid metabolism is altered.

Efforts to design drugs that engage PP2A and importins would likely be met with skepticism due to an expectation of high toxicity to non-transformed cells. However, published studies demonstrate that SH-BC-893 is well tolerated at the effective dose (Kim et al, 2016; Jayashankar et al, 2021; Finicle et al, 2023). At 120 mg/kg administered orally, SH-BC-893 reduces nuclear MYC levels in tumors (Appendix Fig. S8B,C), disrupts PP2A-dependent trafficking (Kim et al, 2016; Finicle et al, 2023) and inhibits autochthonous prostate tumor growth by 82% (Kim et al, 2016). Importantly, this treatment did not lead to bone marrow suppression or disrupt blood chemistry, either acutely or after 11 weeks of continuous treatment (Kim et al, 2016; Finicle et al, 2023). Notably, PP2A activation and importin inhibition are observed at the same

concentrations of SH-BC-893 (Fig. 4). Thus, where PP2A-dependent effects occur in vivo (Jayashankar et al, 2021; Finicle et al, 2023) importins are also likely inhibited. Like yeast that respond to phytosphingosine with an adaptive growth arrest (Chung et al, 2001; Dickson, 2008), non-transformed cells appear to withstand the balanced decrease in nutrient supply and demand that would follow from dual PP2A activation and importin inhibition; evolution may have selected dual PP2A activation and importin inhibition as safe and effective mechanism for homeostatic growth control. In fact, SH-BC-893 protects mice from the metabolic dysfunction that drives diet-induced obesity by preventing endoplasmic reticulum stress and improving mitochondrial function in mice consuming a high fat diet (Jayashankar et al, 2021). Voluntary wheel running, a holistic and sensitive measure of mouse health (Häger et al, 2018), is identical in mice treated with SH-BC-893 and in vehicle controls (Jayashankar et al, 2021). SH-BC-893's 10 h half-life and the incomplete inhibition of pro-growth pathways likely contribute to its tolerability. By analogy, starvation is lethal, but calorie restriction and every other day fasting both have health benefits. These published studies demonstrating that SH-BC-893 is well tolerated at effective concentrations (Kim et al, 2016; Jayashankar et al, 2021; Finicle et al, 2023) offer hope that drugs that engage PPP2R1A and these importins could have an acceptable therapeutic index in lethal diseases where there is an unmet need for new therapeutic approaches.

Attempts to modulate sphingolipid levels for cancer therapy have thus far been unsuccessful, most likely because the network of enzymes that metabolize endogenous sphingolipids make it very difficult to stably elevate the levels of specific sphingolipid species. For example, specific SK1 inhibitors developed as drug candidates dramatically reduced S1P levels but did not slow cancer cell proliferation in vitro or limit tumor growth in vivo (Schnute et al, 2012; Rex et al, 2013). Consistent with the inhibitor data, knocking down SK1 or SK2 with RNAi also did not slow the growth of >20 human cancer cell lines (Rex et al, 2013). These two studies (Schnute et al, 2012; Rex et al, 2013) undermined the hypothesis that S1P is a critical driver of neoplastic growth. However, SK1 inhibition is not the best therapeutic strategy to elevate sphingosine levels. Although the SK1 inhibitor PF-543 increased endogenous sphingosine levels 3-fold in HeLa cells at 1 h (Fig. 6G), chronic SK1 inhibition will trigger compensatory changes that reverse sphingosine accumulation (Appendix Fig. S10A). Sphingosine levels were not reported in cells treated with SK1 inhibitors in (Rex et al, 2013). In (Schnute et al, 2012), 7 days of treatment with PF-543 decreased S1P levels tenfold but only increased sphingosine levels twofold in the single cell line evaluated, a head and neck cancer cell line with exceptionally high SK1 activity. When conversion to S1P is blocked by inhibiting SK1, these head and neck cancer cells detoxified sphingosine by converting it to ceramide (Schnute et al, 2012) (Fig. 5C) which is readily transformed into other sphingolipids that lack growth suppressive effects (Spassieva et al, 2007; Hannun and Obeid, 2018). Constrained sphingosine analogs like SH-BC-893 that are resistant to metabolism (Chen et al, 2016) and that bind PPP2R1A and importins at lower concentrations than sphingosine (Fig. 5A,B) are thus better positioned to become effective therapeutics than compounds designed to alter endogenous sphingolipid levels. The availability of target engagement assays (Fig. 1E–H) and crosslinking data (Fig. 2A–C; Appendix Figs. S5 and S6) should facilitate the optimization of these

molecules for both potency and ADME properties even though their unfolding effects on the targets have so far precluded attempts to fine map their interaction sites using crystallization or NMR. In sum, this work suggests a new strategy to leverage the anti-neoplastic actions of sphingolipids therapeutically.

Parallel PP2A activation and importin inhibition could delay or circumvent the drug resistance that makes cancer a lethal disease, providing a strong rationale for optimizing the pharmaceutical properties of sphingosine-like compounds. Tumor heterogeneity and adaptive genetic and epigenetic changes mean that many cancer therapies lose efficacy within months (Vasan et al, 2019; Marine et al, 2020; Bergholz and Zhao, 2021; Mullard, 2020; Plana et al, 2022). The standard approach to overcome drug resistance is to use combinations of drugs to simultaneously target primary and compensatory pathways and/or heterogeneous tumor drivers. However, divergent pharmacological properties and increased toxicity can complicate the sustained and simultaneous engagement of multiple targets by different drugs. Polypharmacology, inhibiting multiple targets with a single compound, has shown therapeutic promise, but has so far been limited to inhibiting multiple kinases in parallel (Knight et al, 2010; Vincent et al, 2022). Here, we demonstrate the feasibility of engaging multiple difficult to drug, functionally divergent targets, PPP2R1A and four importins, with a single compound. Engaging these targets inhibits many high value oncogenic proteins that are themselves challenging to drug (JUN, YAP, MYC, and AR-V7) and does so under conditions where single pathway compounds are not effective (Fig. 4). These effects on multiple oncogenic pathways would make sphingosine-like compounds active in heterogeneous tumors and drug-resistant tumors. JUN, YAP, MYC, hnRNPA1, and AR-V7 promote resistance to standard of care therapies like enzalutamide, abiraterone, and docetaxel (Antonarakis et al, 2014; Sharp et al, 2019; Riedel et al, 2021; Kuser-Abali et al, 2015; Rodriguez-Bravo et al, 2018; Ouyang et al, 2008; Lee et al, 2021; Nguyen et al, 2015; Kau et al, 2004; Nadiminty et al, 2015). Thus, compounds that target PPP2R1A and importins could be effective in patients that have failed these treatments. Given that PPP2R1A and these importins are ubiquitously expressed and that JUN, YAP, MYC, and NF-κB drive many different types of cancer (Ji et al, 2019; Taniguchi and Karin, 2018; Chen et al, 2018a; Park et al, 2020; Zanconato et al, 2019), sphingosine-like drugs should be effective across cancer classes, not just against prostate tumors. In conclusion, compounds that target PPP2R1A and importins could show therapeutic potential in treatment-resistant tumors and across cancer classes.

# Methods

### Reagents and tools table

| Reagent/resource | Reference or source | Identifier or catalog number |
|---|---|---|
| **Experimental models** | | |
| 293T | ATCC | CRL-3216 |
| 293T-AID-2R1A | This study | |
| 22Rv1 | ATCC | CRL-2505 |
| ACER1-TET-ON HeLa | PMCID: PMC2630785 | |
| FL5.12 cells | Craig Thompson, MSKCC | |
| HeLa | ATCC | CRM-CCL-2 |

| Reagent/resource | Reference or source | Identifier or catalog number |
|---|---|---|
| LNCaP clone FGC | ATCC | CRL-1740 |
| mPCE | PMCID: PMC5096903 | Generated from an autochthonous tumor in *Pten^flox/flox*;*p53^flox/flox*;*PB-Cre4* mouse |
| mPCE-doxi-Kpnb1 | This study | |
| MYC-CaP | ATCC | CRL-3255 |
| SK1 wild-type and knockout MEFs | Ashley Snider, University of Arizona | |
| **Recombinant DNA** | | |
| pX330–Bbs1-PITCh | Addgene | plasmid #127875 |
| pN-PITCh-H3FAID | Addgene | plasmid #127884 |
| pReceiver-B31-mPPP2R1A-6xHIS | Genocopeia | custom order |
| pET-16b-10xHIS-Kpnb1-442 | Genscript | custom order |
| pEGFP-C1-AR V7 | Addgene | plasmid # 86856 |
| Kpnb1-MYC-GLAG expression | Origene | cat# RC200659 |
| pRev-TRE | Clontech (available from NovoPro) | V007373 |
| **Antibodies** | | |
| α-actinin | Cell Signaling Technology | 69758 |
| Androgen receptor | Santa Cruz | sc-7305 |
| FLAG | Sigma | F1084 |
| GAPDH | Cell Signaling Technology | 50190710 |
| hnRNPA1 | Cell Signaling Technology | 8443S |
| JUN | Cell Signaling Technology | 9165S |
| Kpnb1 | ThermoFisher | MA3070 |
| MYC | Cell Signaling Technology | 5605S |
| NF-κB p65 | Cell Signaling Technology | 8242S |
| tubulin | Sigma | T4026 |
| YAP | Santa Cruz | sc-101199 |
| **Oligonucleotides and other sequence-based reagents** | | |
| AID-2R1A sgRNA FOR primer | Eurofins Scientific | 5'-caccgAAGGGACGGAGCCAAGATGG-3' |
| AID-2R1A sgRNA REV primer | Eurofins Scientific | 5'-aaacCCATCTTGGCTCCGTCCCTT-3' |
| AID-2R1A Gibson FOR primer | Eurofins Scientific | 5'-CGCGTTACATAGCATCGTACGCGTACGTGTTTGGGGAAAGGGACGGAGCCAAGAGCCACCATGGTGAGCAAGGG-3' |
| AID-2R1A Gibson REV primer | Eurofins Scientific | 5'-GCATCGTACGCGTACGTGTTTGGTCGTCGCCGTCGGCCGCCGCTTTATACATCCTCAAATCGATTTTCCTCAAGT-3' |
| AID-2R1A genomic validation FOR primer 1 | Sigma | 5'-CGACCAATGGCAAATCGG-3' |
| AID-2R1A genomic validation REV primer 1 | Sigma | 5'-GGAGATAAGAGACGCACG-3' |
| AID-2R1A genomic validation FOR primer 2 | Sigma | 5'-CGACCAATGGCAAATCGG-3' |
| AID-2R1A genomic validation REV primer 2 | Sigma | 5'-GTTTACGTCGCCGTCCAG-3' |
| RT-PCR primers GAPDH (human) | Sigma | For: 5'-ATGGGGAAGGTGAAGGTCG-3' Rev: 5'-GGGGTCATTGATGGCAACAATA-3' |
| RT-PCR primers p21 (human) | Sigma | For: 5'-AGGTGGACCTGGAGACTCTCAG-3' Rev: 5'-TCCTCTTGGAGAAGATCAGCCG-3' |
| RT-PCR primers p27 (human) | Sigma | For: 5'-ATAAGGAAGCGACCTGCAACCG-3' Rev: 5'-TTCTTGGGCGTCTGCTCCACAG-3' |

| Reagent/resource | Reference or source | Identifier or catalog number |
|---|---|---|
| RT-PCR primers GAS1 (human) | Sigma | For: 5'-ATCTGCGAGTCGGTCAAGGAGA-3' Rev: 5'-CTGCTCGTCAT CGTAGTCCTCA-3' |
| RT-PCR primers ODC1 (human) | Sigma | For: 5'-TGTAGGAAGCGGCTGTAC-3' Rev: 5'-GCTATGATTCTCACTC CAGAG-3' |
| RT-PCR primers PAICS (human) | Sigma | For: 5'-GCAGCAAATTGAGCTGATCC-3' Rev: 5'-ACTGGGGAGTTCAGGATGTG-3' |
| RT-PCR primers LDHA (human) | Sigma | For: 5'-TTGACCTACGTGGC TTGGAAG-3' Rev: 5'-GGTAACGGA ATCGGGCTGAAT-3' |
| RT-PCR primers CCND2 (human) | Sigma | For: 5'-GTAAAGACAGCC TTGACTCAAGCAT-3' Rev: 5'- CGGTGAAGTCCGCCAGC-3' |
| RT-PCR primers DNMT1 (human) | Sigma | For: 5'-AGGCGGCTCAAAGAT TTGGAA-3' Rev: 5'- GCAGAAATTCGTGCA AGAGATTC-3' |
| RT-PCR primers FKBP5 (human) | Sigma | For: 5'-GCAACAGTAGAAA TCCACCTG-3' Rev: 5'-CTCAGAGCTTTGTCAATTCC-3' |
| RT-PCR primers DEPTOR (human) | Sigma | For: 5'-ACTGGCTGGTTCA GGAAGGTGA-3' Rev: 5'-GCTGTCCACA AATGGGTGCTTG-3' |
| RT-PCR primers TMPRSS2 (human) | Sigma | For: 5'-GTGATGGTATTCA CGGACTGG-3' Rev: 5'-CAGCCCCATTGT TTTCTTGTA-3' |
| RT-PCR primers KLK3 (human) | Sigma | For: 5'-ATATCGTAGAGCGGGTGTGG-3' Rev: 5'-CATCAGGAACAA AAGCGTGA-3' |
| RT-PCR primers SK1 (mouse) | Sigma | For: 5'-TGTCACCCATGAACCTGC TGTCCCTGCACA-3' Rev: 5'-AGAAGGCACTGGCTCCTCC AGAGGAACAAG-3' |
| RT-PCR primers RPS16 (mouse) | Sigma | For: 5'-CACTGCAAACGGGGAAATGG-3' Rev: 5'-CACCAGCAAATCGCTCCTTG-3' |
| **Chemicals, enzymes and other reagents** | | |
| 3-ketosphinganine | Cayman Chemicals | 24380 |
| Azide agarose beads | Jena Bioscience | CLK-1038-2 |
| BCA assay | Fisher Scientific | PI23224 |
| Bovine pituitary extract | Fisher Scientific | 13-028-014 |
| Bovine insulin | Sigma | I0516-5ML |
| C2-ceramide (d18:1/2:0) | Cayman Chemicals | 62510 |
| C6-ceramide (d18:1/6:0) | Cayman Chemicals | 62525 |
| C16-ceramide (d18:1/16:0) | Cayman Chemicals | 10681 |
| ciprofloxacin | Sigma | 17850-5G-F |
| Citrate synthase protein | Sigma | C3260-200UN |
| Dihydrotestosterone | Sigma | 14z31573 |
| Dimethylsphingosine (d18:1) | Cayman Chemicals | 62575 |
| DMEM | Fisher Scientific | MT10017CV |
| epidermal growth factor, human | Fisher Scientific | PHG0311 |
| epoxomicin | Apex BioTechnology | A2606 |
| FTY720 | Cayman Chemicals | 10006292 |
| FTY720-biotin | Cayman Chemicals | 13254 |
| HEPES | Fisher Scientific | 25-060-Cl |
| HisPur Ni-NTA Resin | ThermoFisher | PI88221 |
| IL-3 (recombinant mouse) | Fisher Scientific | 50-170-401 |
| importazole | Selleck Chemicals | S8446 |
| Myoglobin (recombinant) | Sigma | M0630 |
| Penicillin/streptomycin | ThermoFisher | MT30002CI |
| perphenazine | Sigma | P6402 |
| PF-543 | Cayman Chemicals | 17034 |
| phytosphingosine | Cayman Chemicals | 20217 |

| Reagent/resource | Reference or source | Identifier or catalog number |
|---|---|---|
| RPMI 1640 | Fisher Scientific | A1049101 |
| safingol | Cayman Chemicals | 18624 |
| Sphingosine d12:1 | Cayman Chemicals | 26893 |
| Sphingosine d18:1 4E | Cayman Chemicals | 10007907 |
| Sphingosine d18:1 14Z | Cayman Chemicals | 27019 |
| Sphingosine d20:1 | Cayman Chemicals | 10007903 |
| Sphingosine d17:1 | Cayman Chemical | 10007902 |
| D-erythro-sphingosine d18:1 phosphate | Cayman Chemicals | 62570 |
| D-erythro-sphingosine d17:1-phosphate | Cayman Chemicals | 22498 |
| SYPRO orange | Millipore Sigma | S5692 |
| TNFα | Biolegend | 570102 |
| **Software** | | |
| ImageJ | https://imagej.net/downloads | |
| ProteoDiscoverer | ThermoFisher | |
| MaxQuant | https://www.maxquant.org/ | |
| R | https://cran.r-project.org/ | |
| Prism | GraphPad | |
| **Other** | | |
| BL-21 E. coli | NEB | C2527H |
| Arctic Express E. coli | Agilent | 230192 |
| Ultracel-30 Millipore Amicons, 30kD NMWL | Fisher Scientific | UFC903024 |
| Pierce trade Slide-A-Lyzer trade G2 Dialysis Cassettes | VWR | 87717 |

## Cell lines and cell culture

All cell lines were maintained at 37 °C in 5% $CO_2$. FL5.12 were obtained from Craig Thompson (Memorial Sloan Kettering Cancer Center) and mPCE cells were derived in-house from prostate tumor tissue from a *Pten*<sup>flox/flox</sup>; *p53*<sup>flox/flox</sup>; *PB-Cre4* mouse. SK1 WT and KO MEFs were provided by Ashley J. Snider (University of Arizona). 22Rv1, MYC-CaP, LNCaP, 293T, and HeLa cells were obtained from the ATCC. FL5.12 cells were cultured in RPMI 1640 medium supplemented with 10% fetal bovine serum (FBS), 10 mM HEPES, 55 μM β-mercaptoethanol, antibiotics, 2 mM L-glutamine, and 500 pg/ml murine rIL-3. mPCE were maintained in DMEM with 10% FBS, 25 μg/ml bovine pituitary extract, 5 μg/ml bovine insulin and 6 ng/ml recombinant human epidermal growth factor. For triple SILAC (stable isotope labeling by amino acids in cell culture), FL5.12 cells were grown in RPMI 1640 media supplemented with SILAC amino acids (SILANTES GmBH), 10% dialyzed FBS, 10 mM HEPES, 55 μM β-mercaptoethanol, antibiotics, 2 mM L-glutamine, and 500 pg/ml rIL-3. mPCE cells were grown in DMEM supplemented with 10% dialyzed FBS, SILAC amino acids, 25 μg/ml bovine pituitary extract, 5 μg/ml bovine insulin, and 6 ng/ml recombinant human epidermal growth factor. Both cell lines were grown for at least 9 doublings in SILAC medium prior to use in proteomics experiments. 22Rv1 and LNCaP were maintained in RPMI supplemented with 10% FBS. MYC-CaP, HeLa, and MEFs were cultured in DMEM with 10% FBS. Cell volume was determined with a Coulter counter (Becton

Dickinson) and cellular protein content determined using a BCA Assay (ThermoFisher).

To generate 293T-AID-2R1A cells expressing N-terminally AID-tagged PPP2R1A, a microhomology repair strategy was followed (Natsume et al, 2016). Briefly, guide RNAs were designed using the tool provided at https://portals.broadinstitute.org/gpp/public/analysis-tools/sgrnadesign. Oligos were synthesized at Euro-fins Scientific with a 20-bp gRNA sequence containing a 5' overhang CACC and 3' overhang CAAA (FOR primer caccgAAGGGACGGAGCCAAGATGG and REV primer aaacCCATCTTGGCTCCGTCCCTT) to facilitate cloning into the Bbs1 site of pX330–Bbs1-PITCh (Addgene plasmid #127875). Microhomologies on the pN-PITCh-H3FAID tagging vector (Addgene plasmid #127884) were changed to match the PPP2R1A locus by PCR and cloned into the vector backbone using Gibson assembly (FOR primer: CCGCGTTACATAGCATCGTACGCGTA CGTGTTTGGGGAAAGGGACGGAGCCAAGAGCCACCATG GTGAGCAAGGG; REV primer: GCATCGTACGCGTACGTG TTTGGTCGTCGCCGTCGGCCGCCGCTTTATACATCCTCAAA TCGATTTTCCTCAAGT). Five µg of pX330 and 2.5 µg of pN-PITCh tagging vector were used to transfect 293T cells that stably expressed TIR1. Cells were selected with 1 µg/ml puromycin 72 h post-transfection. Clonal cell lines were generated by limiting dilution and the degradation of HF-AID-tagged PPP2R1A in response to 500 µM NAA monitored by immunoblotting or immunofluorescence microscopy. Recombination was also verified by PCR using the strategy shown in Appendix Fig. S8B,C. To generate mPCE-doxi-KPNB1 cells, human KPNB1-MYC-FLAG (Origene) was cloned into pRevTRE (Clontech) and introduced by transduction into mPCE cells expressing rtTA.

Cell lines were screened for *Mycoplasma* contamination at least monthly using the protocol outlined in (Uphoff and Drexler, 2014) and in DAPI images evaluated for contamination every time microscopy was performed. Cells found to be contaminated with *Mycoplasma* were discarded along with all data collected in these lines back to the last negative *Mycoplasma* test. ACER1-TET-ON HeLa cells were cured of *Mycoplasma* by treating with ciprofloxacin for 6 weeks (resolution of infection confirmed by PCR). Thus, all data presented is from cell lines confirmed to be *Mycoplasma*-free. Cell viability was measured using vital dye exclusion and flow cytometry (FL5.12) or crystal violet staining (mPCE). IC50 values were calculated using GraphPad Prism.

## Proteomics

LC-MS/MS analyses for chemoproteomics and nuclear proteomics were performed on a Q-Exactive HF or an Orbitrap tribrid Fusion mass spectrometer using homemade capillary LC columns (18 cm length, 150 µm ID, 360 µm OD) and a Thermo EASY nLC system coupled to an ESI (Electrospray Ionization) source. Capillary LC columns were packed with C18 Jupiter 3 µm particles (Phenomenex, Torrance, CA) at 1000 psi. Samples were directly injected on the LC-MS/MS system using a flow rate of 0.6 µL/min with a linear gradient of 5-35% aqueous acetonitrile (ACN with 0.2% formic acid) over 120 min. MS spectra were acquired with a resolution of 60,000 using a lock mass ($m/z$: 445.120025) followed by up to 20 MS/MS data dependent scans at a resolution of 15,000 on the most intense ions using high energy dissociation (HCD). AGC target values for MS and MS/MS scans were set to $1 \times 10^6$ (max fill time

100 ms) and $5 \times 10^5$ (max fill time 200 ms), respectively. The precursor isolation window was set to $m/z$ 1.6 with a HCD normalized collision energy of 25. The dynamic exclusion window was set to 20 s.

Intact mass analyses were performed on a Thermo Q-Exactive Biopharma instrument. A C4 pre-column was used to desalt and separate intact proteins. Proteins were eluted from the pre-column using a linear gradient of 5–40% aqueous ACN (0.2% formic acid) over 20 min with a flow rate of 2 µl/min using the Ion Max Atmospheric Pressure Ionization (API) source configured with the heated electrospray ionization probe. Intact mass spectra were acquired at a resolution of 15,000 at $m/z = 200$ in full scan mode. Following each injection, the pre-column was washed with 80% aqueous ACN (0.2% formic acid) for 2 min and equilibrated with 5% ACN for 10 min. Samples were analyzed using ProteoDiscoverer, spectra were deconvoluted using a target mass of 65 kD for PPP2R1A and 44 kD for KPNB1.1 (aa 1–442).

## Proteomics data analysis

All chemoproteomics data were analyzed using MaxQuant. MaxQuant and Andromeda search parameters are specified in Dataset EV1. ProteinGroups.txt was used for further data analysis in R to extract significantly enriched proteins. The mass spectrometry proteomics data have been deposited to the ProteomeXchange Consortium via the PRIDE repository with the dataset identifier PXD036429. Selection criteria for proteins represented on heatmaps are described in the corresponding figure legends. Heatmaps represent either the relative enrichment (active/inactive compound, Fig. 1C) or direct enrichment (active compound/control beads, Appendix Fig. S1G). To identify significantly regulated features in Fig. 3A, we applied a curved significance threshold on the volcano plot, which combines both fold-change magnitude and $P$ value significance. This threshold represents a false discovery rate (FDR)-adjusted significance boundary, ensuring that only features with both high effect size (e.g., $|\log_2FC|$) and strong statistical support ($-\log_{10}(P$ value)) are highlighted. The curved boundary accounts for the fact that small fold changes can still be significant with very low $P$ values, and conversely, large fold changes must also show statistical reliability. This method reduces false positives by avoiding arbitrary cutoffs and better reflects the trade-off between fold change and variance (Tyanova et al, 2016).

## Sample preparation for chemoproteomics

To prepare beads decorated with immobilized sphingosine-like compounds, azide agarose beads (10-20 µmol azide/ml bead slurry, Jena Bioscience, Cat. #CLK-1038-2) were washed twice with 1 ml crosslinking buffer (ammonium acetate ($NH_4CH_3CO_2$, pH 4), 5 mM $CuSO_4$, 1 mM tris(3-hydroxypropyltriazolylmethyl)amine (THTPA), 20 mM sodium ascorbate) by centrifugation at 3000 × $g$ for 30 s and 100 µl of bead slurry suspended in a final volume of 300 µl. Alkyne-containing compounds solubilized in DMSO were added to the bead suspension (1:200 compound:azide assuming 15 µmol azide/ml bead slurry or 75 nmol compound/ml in a reaction with 100 µl of bead slurry) and incubated 14 h at 37 °C. Control, unconjugated beads were incubated with the DMSO vehicle alone. Crosslink yields were quantified using an Agilent LC-MSD system equipped with atmospheric pressure chemical

ionization. A linear gradient elution (mobile phase A: 0.2% TFA in water, mobile phase B: 0.2% TFA in methanol) of 5 to 100% B in 2 min and a Kinetex-C18 column were used to determine the fraction of free compound remaining after crosslinking; crosslinking efficiency was >99% in all reactions. Beads were washed five times with 1 ml PBS and then 5 times with chemoproteomics lysis buffer (150 mM NaCl, 0.5% NP-40, 25 mM Tris pH 7.4) prior to storage at 4 °C in chemoproteomics lysis buffer at 15 µmol compound/ml slurry (assuming 100% crosslink yield).

Heavy, medium, or light SILAC-labeled FL5.12 or mPCE cells were lysed by vortexing in ice-cold chemoproteomics lysis buffer for 10 min. The resulting homogenate was spun at 13,000 × $g$ for 15 min at 4 °C and the supernatant analyzed for protein content using a Bradford assay. Lysates were adjusted to equivalent protein concentrations across replicates, incubated with beads for 18 h at 4 °C (50 µl of functionalized bead slurry with 500 µl/700 µg of lysate), washed three times with 1 ml ice-cold chemoproteomics lysis buffer spinning at 3000 × $g$ for 1 min, after which beads that had been incubated with heavy, medium, or light protein extracts combined and washed five times with 1 ml ice-cold PBS and twice with 300 µl of 50 mM ammonium bicarbonate to remove excess salts. On-bead digestion with trypsin (5 µg/sample) was performed in 50 mM ammonium bicarbonate for 18 h at 37 °C with gentle agitation prior to MS analysis.

## UV crosslinking and intact mass analysis

SH-BC-893-diazirine was incubated with 2 µM purified PPP2R1A or KPNB1 at concentrations ranging from 0–20 µM in 100 µl in a 96-well plate on ice; the bottom of the plate was covered with aluminum foil. Reactions were performed in wells A1-A6, B1-B6, and C1-C6 to keep UV intensity constant; sample positions were randomized in each replicate. A 20 W UV lamp emitting at 365 nm (American DJ Black-24BLB) was placed 1 cm away from the plate for 10 min after which 20 µl of each sample was immediately transferred to mass spectrometry vials and kept at 4 °C in the dark until preparation for injection. Samples were incubated with 2 µg trypsin for 4 h then dried in a SpeedVac centrifuge. Peptides were resuspended in 4% formic acid and approximately 20 fmol of peptides injected onto the LC-MS/MS system as described above after desalting. Samples were analyzed using MaxQuant. The function "custom modification" in MaxQuant was used to build the modifications "SH-BC-893" or "SH-BC-893-NAc" which were used to detect peptides crosslinked to the diazirine-tagged small molecules. Manual inspection of MS/MS spectra was performed following Andromeda searches to ensure crosslinked peptides were correctly identified.

## Nuclear protein extraction for proteomics

Approximately 20 million light, medium, or heavy SILAC-labeled mPCE cells were treated with either DMSO or 5 µM SH-BC-893 in DMSO for 6 h, trypsinized, and then lysed with 1 ml of hypotonic lysis buffer (10 mM Tris-HCl pH 7.4, 10 mM NaCl, 3 mM MgCl₂, 0.3% v/v NP-40, 10% v/v glycerol) for 10 min on ice then vortexed quickly. Complete cell lysis was confirmed by light microscopy. Nuclei were washed three times with isotonic wash buffer (0.25 M sucrose, 5 mM MgCl₂, 10 mM Tris-HCl, pH 7.4) and subsequently lysed in 8 M urea in 15 mM Tris, pH 8. Proteins were quantified

using a Bradford assay then reduced and alkylated using tris(2-carboxyethyl)phosphine (TCEP, 2 µM) and chloroacetamide (CAA, 5 µM) prior to dilution to 1 M urea with 15 mM Tris, pH 8. Tryptic digestion of protein extracts (50:1, substrate:enzyme, w:w) was performed overnight at 37 °C with gentle shaking. To facilitate quantitation of non-histone proteins, peptides were fractionated into 20 samples using high pH reverse phase chromatography on a Waters XBridge BEH130 C18 column (250 ×4.6 mm i.d.) using an Agilent 1100 HPLC system and the mobile phases A (50 mM NH₄HCO₃, pH 7.5) and B (50 mM NH₄HCO₃, pH 7.5 in 40% ACN (v/v)) and a gradient of 5% to 100% B over 60 min at a flow rate of 800 µl/min. The column was washed with 90% ACN (v/v) for 5 min and equilibrated with mobile phase A for 20 min prior to each injection. Approximately 2 mg of tryptic digest was injected per biological replicate. Fractions were dried in a SpeedVac centrifuge prior to LC-MS/MS analysis.

## Recombinant protein production

Mouse PPP2R1A-6XHis (full length, C-terminally tagged) and 10X-His-KPNB1.1 (amino acids 1–442 as in (Bayliss et al, 2000), N-terminally tagged, codon-optimized) in bacterial expression vectors (pReceiver-B31 or pET-16b, respectively) were purchased from Genocopeia or Genscript, respectively. BL-21 (NEB) or Artic Express (Agilent) *E. coli* were used for protein expression. Bacteria were grown at 37 °C in LB supplemented with 100 µg/ml ampicillin until 1 mM IPTG was added at an OD of 1. Cultures were transferred to 16 °C and bacteria were harvested ~18 h later. Bacteria were lysed using five cycles of 1 min of sonication. Target proteins were purified on Ni-NTA beads (ThermoFisher, 1 L of culture to 5 ml of bead slurry) pre-equilibrated with buffer A (50 mM KH₂PO₄, pH 7.4, 150 mM NaCl, 1 mM DTT). Samples were gently rotated at 4 °C for 30 min then washed 10–15 times with 10 ml of buffer B (50 mM KH₂PO₄, pH 7.4, 150 mM NaCl, 1 mM DTT, and 20 mM imidazole), and protein eluted in 5 ml of buffer C (50 mM KH₂PO₄, pH 7.4, 150 mM NaCl, 1 mM DTT, and 250 mM imidazole) until A280 reached baseline. Purified proteins were concentrated and imidazole removed with 2 × 15 ml buffer A washes in a centricon (Millipore UFC903024), quantified using A280, and stored at 4 °C (short term) or −80 °C (long term) in buffer A. When required, buffer was exchanged using a PD-10 desalting column (Sigma, GE17-0851-01). Recombinant citrate synthase protein was purchased from Sigma (C3260-200UN).

## Thermal shift assays and circular dichroism

Circular dichroism (CD) spectra of PPP2R1A and KPNB1 N-terminus (purified in-house) and purchased myoglobin (Sigma cat# M0630) were measured on a Jasco J-810 spectropolarimeter at 20 °C with 5 µM protein unless otherwise noted. Protein was diluted and transferred to a 1 mm path length quartz cuvette and the far-UV spectrum recorded from 195 to 260 nm in triplicate; spectra were baseline-corrected for buffer alone. To measure effects of SH-BC-893 or 893-ketone on secondary structure, increasing concentrations of compound were added to the same sample outside of the cuvette following a baseline measurement. Fractional helicity at 208 nm was calculated by dividing peak height with compound by peak height at baseline to allow comparison of multiple compounds and buffer conditions. Differential scanning

fluorimetry (DSF) assays was performed in triplicate using 5 μM protein and 1/1000 volume SYPRO Orange (Sigma cat #S5692) on a Stratagene Mx3005P real-time PCR machine (Agilent) or a QuantStudio3 (ThermoFisher) as described in (Vivoli et al, 2014). Single traces not matching the others in a set or first derivative traces starting below 0 were omitted from the analysis. Where indicated, ligand was removed by dialyzing O/N at 4 °C using 2000 MWCO Slide-A-Lyzer Dialysis G2 cassettes (Thermo Scientific cat# 87717) with buffer exchanges at 2 and 4 h. All DSF and CD shown was conducted in DSF buffer (10 mM $NaH_2PO_4$ pH 6.0, 25 mM NaCl, 1 mM β-mercaptoethanol).

## Computational modeling structure preparation

First, the protein structure of KPNB1 was prepared for ligand docking from PDB ID 1F59 chain A (Bhardwaj and Cingolani, 2010) with residue charges titrated by the H + + Server (Gordon et al, 2005) to a pH of about 7. Next, eight conserved interaction sites discovered by UV crosslinking experiments were used as initial input coordinates for SH-BC-893 docking using the center of mass of the following residues: E19, K23, E31, R42, K183, E185, D379, and R381. The SMILES code for SH-BC-893, (2S,3R)-3-(4-octylphenyl)pyrrolidin-2-yl)methanol C([C@H]1[C@H](C2 = CC = C(CCCCCCCC)C = C2)CCN1)O was input into the System Builder package (GitHub DOI: 10.5281/zenodo.6658415, Author: Mary Pitman, Year: 2021) to perform docking. System Builder docked ligands with OpenEye Toolkits 2021.2.0 to perform multiconformer docking onto an OEReceptor generated from 3LWW (Bhardwaj and Cingolani, 2010). We found five representative docked binding poses from the eight initial guesses for binding locations derived from UV crosslinking.

## All-atom computational modeling

The five docked complexes were prepared as starting structures for simulation to determine which locations displayed stable ligand binding. Computational studies employed all-atom molecular dynamics simulations with the GROMACS 2021.2 software suite (Abraham et al, 2015). The systems were parameterized with force fields amber99SB*-ILDNmut (Gapsys et al, 2015), the General AMBER Force Field 2 (GAFF2) for ligands, and the TIP3P water model. Ligand parameters were generated for simulation using AmberTools with docked ligand coordinates to generate parameter, topology, and coordinate files in format. ParmEd (Shirts et al, 2017) was then used to convert the ligand input files to GROMACS format. Bash scripts were written to perform the following simulation protocols over the five simulation systems, and hence the following methods were implemented identically across simulations. Initial complex GRO files were generated with the pdb2gmx command and periodic boundary conditions were employed. The boundaries of the simulations were set to a triclinic box with a minimum distance of 1.0 nm to the periodic boundary, where the complex was aligned in the box by the protein's principal axes. Index files and heavy atom restraints were generated for the ligands with force constants of $1000 \, kJ \, mol^{-1} \, nm^{-2}$ for the $x$, $y$, and $z$ axes. The complex was then solvated and counter ions of Na+ and Cl− were introduced to neutralize the complex and to model an ionic concentration of 150 mM NaCl.

Complexes were energy minimized using steepest descent to a maximum energy of 10 kJ/mol. For electrostatics, we employed the Particle Mesh Ewald method with the Verlet cut-off scheme. The long range electrostatic and Van der Waals cut-off distance for non-bonded interactions was set to 1.0 nm. Following minimization, system equilibrations were performed with covalent bonds to hydrogens constrained with the LINCS algorithm. First, NVT equilibration was carried out with both the ligand and protein restrained at $1000 \, kJ \, mol^{-1} \, nm^{-2}$. The complex was treated as a group for temperature protein and ligand coupling. The systems were heated to 300 K for 100 ps. The modified Berendsen thermostat was used with a time constant of 0.1 ps with two coupling groups, (1) protein and ligand and (2) water and ions. The system pressure was then equilibrated for 1000 ps in the isothermal-isobaric ensemble at 300 K, 1.0 bar pressure with Berendsen pressure coupling, isotropic scaling of box vectors, and a coupling time constant of 2.0 ps. Temperature and pressure were checked after each equilibration step to check that the equilibrated intensive variables reached stable oscillation about the reference temperature and pressure.

Each production simulation was run for 5 ns at 300 Kelvin with a timestep of 2 fs. The simulation temperature was V-rescaled using the modified Berendsen thermostat (Lemak and Balabaev, 1994) with the two coupling groups as defined in NVT equilibration with time constant of 0.1 ps. The Parrinello–Rahman barostat (Bussi et al, 2009) was used to maintain a pressure of 1.0 bar with a coupling time constant of 2.0 ps and isotropic box vector scaling. At the end of the production runs, coordinates were recentered, rewrapped, and rotational and translational motions of the protein backbone were fit. To quantify the extent of ligand motion from the five potential binding sites, we calculated the root-mean-square deviation (RMSD) of SH-BC-893 heavy atoms with respect to the protein backbone and plotted the RMSD over simulation timespan. Further, we visually inspected the binding mode throughout the simulations and calculated the number of stable hydrogen bonds.

## Microscopy

For immunofluorescence microscopy, cells were plated in eight-well chamber slides (CellVis cat# C8-1.5H-N) such that they would be 60–75% confluent at the time of treatment 16–24 h later. Cells were treated as indicated in figure legends then washed twice with PBS and fixed with 4% paraformaldehyde for 10 min at RT. Cells were incubated in blocking buffer containing 10% fetal bovine serum and 0.3% saponin for 20 min at 37 °C followed by overnight incubation with primary antibodies at 4 °C. Cells were then washed twice with PBS and incubated with secondary antibodies at RT followed by 5 min incubation with 1 μg/ml DAPI and two washes in PBS. Fluorescence microscopy was performed on a Zeiss LSM 780 confocal microscope using a ×63 oil objective with a 1.7 numerical aperture using Zeiss Zen 2.3 image acquisition software; 16-bit images were acquired. Nuclear protein levels were quantified in single 2 micron sections using ImageJ. Nuclear ROIs were defined in the DAPI image and the mean gray value for each cell recorded for each protein of interest. In each experiment, data was normalized relative to the median value of the control sample to allow comparisons between experiments done on different days. Cells undergoing mitosis or not entirely within the field of view

were excluded from the analysis. The median rather than the mean of the control sample was used for normalization because microscopy data was not normally distributed. At least 100 cells from each of 2 biological replicates was analyzed unless otherwise indicated in the legend. Antibodies used for IF were: KPNB1 (1:250, ThermoFisher cat# MA3070); YAP (1:100, Santa Cruz Biotechnology cat# sc-101199), JUN (1:500, Cell Signaling Technology cat# 9165S), MYC (1:250, Cell Signaling Technology cat# 5605S), AR (1:100, Santa Cruz Biotechnology cat# sc-7305), FLAG (1:100, Sigma cat# F1084), p65 (1:250, Cell Signaling Technology cat# 8242S).

Immunohistochemistry for MYC was performed by the UCI Pathology Core on archived pDKO FFPE tumors from (Kim et al, 2016). Stained tumors from mice treated with vehicle (water) or 120 mg/kg SH-BC-893 PO QD 5–7 days a week for 11 weeks were evaluated on a Nikon Ti2-F inverted epifluorescence microscope equipped with a DS-Fi3 color camera. Eight to 12 non-overlapping fields were acquired; >2500 nuclei from tumors in three different mice per treatment group were quantified. Representative images are shown. Nuclear MYC levels were quantified in RGB images using ImageJ and the approach outlined in (Nguyen and Nguyen, 2013). Nuclear ROIs were defined in the hematoxylin channel and the mean intensity recorded. Intensity of the stained region was subtracted from the maximum intensity of a RGB image (250 units) to generate a reciprocal intensity value that is directly proportional to the level of MYC staining.

## Reagents

Specialty chemicals were obtained from the following sources and suspended in the indicated vehicles: dihydrotestosterone (DHT, Sigma Aldrich cat# 31573, 100 mM stock solution prepared in ethanol), perphenazine (PPZ, Sigma Aldrich cat# P6402, 50 mM stock solution prepared in ethanol), importazole (IPZ, Selleck Chemicals S8446, 50 mM stock prepared in DMSO), D-erythro-Sphingosine C-17 (d17:1) (Cayman Chemical, cat# 10007902, 10 mM stock in ethanol), sphingosine (d18:1) (Cayman Chemical cat# 10007907, 10 mM stock solution prepared in ethanol), sphingosine (d20:1) (Cayman Chemical cat# 10007903, 10 mM stock solution prepared in ethanol), sphingosine (d12:1) (Cayman Chemical cat# 26893, 10 mM stock solution prepared in ethanol), D-erythro-sphingosine C17-phosphate (d17:1) (Cayman Chemical cat# 22498, 1 mM stock solution prepared in 37 °C methanol), 3-keto-sphinganine (d18:0) (Cayman Chemical cat# 24380, 10 mM stock solution prepared in ethanol), C2 ceramide (d18:1/2:0) (Cayman Chemical cat# 62510, 50 mM stock solution prepared in DMSO), C6 ceramide (d18:1/6:0) (Cayman Chemical cat# 62525, 50 mM in DMSO), C16 ceramide (Cayman Chemical). FTY720 (Cayman Chemical cat# 10006292, 5 mM in water), FTY720 phenoxy-biotin (Cayman Chemical cat# 13254, 5 mM in DMSO), phytosphingosine (Cayman Chemical cat# 20217, 100 mM stock solution prepared in ethanol), PF-543 (Cayman Chemical cat# 17034, 10 mM stock solution prepared in DMSO), safingol (Cayman Chemical cat# 18624, 50 mM in DMSO), dimethyl-sphingosine (Cayman Chemical cat# 62575, 50 mM in DMSO), epoxomicin (Apex BioTechnology cat# A2606, 5 mM stock solution prepared in DMSO), TNFα (Biolegend cat# 570102). pEGFP-C1-AR V7 was a gift from Michael Mancini and Marco Marcelli (Addgene plasmid # 86856). SMAP used in Appendix Fig. S3K,L was synthesized as described in (Garsi et al, 2019).

## PCR

For RT-qPCR, RNA was isolated using a Direct-zol RNA Purification Kit (R2051, Zymo Research Corporation). Quantity and integrity of RNA was monitored with a NanoDrop™ 2000c Spectrophotometer (ThermoFisher Scientific). For cDNA synthesis, 1 µg of RNA was added to each reaction using iScript™ cDNA Synthesis Kit (Bio-Rad). Quantitative real-time PCR was carried out using a StepOnePlus™ Real-Time PCR System (ThermoFisher Scientific). RT-qPCR primers are provided in the "Reagents and Tools Table".

## Sphingosine and S1P quantification

Cells were washed twice in ice-cold HPLC grade water with 0.9% NaCl, frozen at −80 °C for 1 h, and then harvested using ice-cold 80% methanol and a cell scraper (1000 µl of 80% methanol per 10-cm dish). The sample was snap frozen in liquid nitrogen, thawed, vortexed for 1 min, then centrifuged at 4 °C and maximum speed in a microcentrifuge. For the experiment shown in Fig. 6, the supernatant was collected, dried under nitrogen, and then the re-solubilized in 80 µl of methanol. After centrifuging at 4 °C and maximum speed for 10 min, 50 µL of clear solution was transferred to an LC-MS vial for injection. For experiments shown in Fig. 5, samples were run directly without concentration. A quadrupole-orbitrap mass spectrometer (Q-Exactive, ThermoFisher Scientific, San Jose, CA) operating in positive ion mode was coupled via electrospray ionization and used to scan from $m/z$ 250 to 1000 at 1 Hz and 140,000 resolutions. LC separation was on an Atlantis T3 Column (2.1 mm × 150 mm, 3 µm particle size, 100 Å pore size; Waters, Milford, MA) using a gradient of solvent A (1 mM ammonium acetate, 35 mM acetic acid in 90:10 water:methanol) and solvent B (1 mM ammonium acetate, 35 mM acetic acid in 98:2 isopropanol:methanol). The flow rate was 150 µL/min. The LC gradient was: 0 min, 25% B; 2 min, 25% B; 5.5 min, 65% B; 12.5 min, 100% B; 16.5 min, 100% B; 17 min, 25% B; 30 min, 25% B. Autosampler temperature was 4 °C, and the injection volume was 3 µL. For Fig. 5, 10 µL of sample was injected. Data were analyzed using the EI-MAVEN software. The concentration was calculated based on the standard calibration curves and normalized by the protein content of each group measured by BCA assay. To account for extraction efficiency, 1 µM d17:1 sphingosine and d17:1 S1P were added as internal standards and the ion count from MS analysis was corrected for variable extraction efficiency; sphingosine was normalized to d17:1 sphingosine and S1P to d17:1 S1P. Data is from three separate 10-cm dishes treated on the same day. Cell volumes were determined using a Coulter counter, and protein content was determined by BCA assay.

## Statistical analysis

In line graphs, mean ± SD is presented unless otherwise indicated in the legends; box plots show medians and quartiles (middle 50% of the data) while whiskers span the 25% lowest and highest values. Experimental data is from ≥3 independent biological replicates (each of which represents the mean of 3–4 technical replicates) except where otherwise indicated. Statistical analysis was performed on biological replicates, not technical replicates. Microscopy quantification was performed on a per cell basis using images from ≥2 independent experiments. Sample size was determined based on prior experience with each assay and to reflect standard practice in the field. No blinding was performed in these studies. No samples were excluded from the

analysis with the exception of dividing cells in nuclear quantifications of microscopy images and in Appendix Fig. S8F as explained in the legend. Statistical analysis was performed using GraphPad Prism software except for proteomics data when the statistical package R was used. All microscopy data was evaluated using the D'Agostino–Pearson omnibus normality test. One-way ANOVA (Kruskal–Wallis) or Welch's t test (do not assume equal variance) was performed as indicated in the figure legends. Corrections for multiple comparisons were made where appropriate using the indicated tests and adjusted $P$ values are reported: ns, not significant, $P \geq 0.05$; *$P < 0.05$; **$P < 0.01$; ***$P < 0.001$. Exact $P$ values are provided in a table in the Appendix.

## Synthesis of diazirines and methods of conjugation

**SH-LS-191**                                                                          **SH-LS-244**

### *tert*-butyl (2*S*,3 *R*)-2-(((*tert*-butyldiphenylsilyl)oxy)methyl)-3-(4-(3-heptyl-*3H*-diazirin-3-yl)phenyl)pyrrolidine-1-carboxylate (SH-LS-244)

Benzylamine (72 µL, 0.66 mmol, 3 eq.) and ZnCl$_2$ (1.5 mg, 0.011 mmol, 0.05 eq.) were added to a solution of **SH-LS-191** (141 mg, 0.22 mmol) in dry toluene (3 ml) containing 4 Å molecular sieves and the resulting mixture was refluxed for 8 h. Afterwards, the reaction was diluted with Et$_2$O (20 ml) and washed with H$_2$O (5 ml) and brine (5 ml). Then, the organic layer was dried over Na$_2$SO$_4$, filtered and concentrated in vacuo, resulting in a pale yellow oil. This intermediate imine was dissolved in dry MeOH (3 ml) and added dropwise to a flame-dried three-neck flask containing liquid ammonia (4 ml ca. condensed by connecting directly a tank to the flask and slowly adding gaseous ammonia), equipped with a cold finger at −78 °C and kept in a cool bath at the same temperature and under an argon atmosphere (balloon). The cool bath was removed and the resulting mixture was stirred for 2 h at ammonia refluxing point, by using the cold finger at −78 °C as condenser. Finally, a solution of hydroxylamine-*O*-sulfonic acid (40 mg, 0.35 mmol, 1.6 eq.) in dry MeOH (2 ml) was added dropwise and the reaction was allowed to slowly reach room temperature and stirred overnight. Afterwards, the mixture was concentrated, H$_2$O (5 ml) was added to the residue and the product was extracted with Et$_2$O (2 × 10 ml). The organic layers were washed with brine (5 ml), dried over MgSO$_4$, filtered and concentrated, resulting in a colorless oil. This intermediate diaziridine was dissolved in dry CH$_2$Cl$_2$ (3 ml) in a flask wrapped in aluminum foil and cooled down to 0 °C, before adding Et$_3$N (92 µL, 0.66 mmol, 3 eq.). The mixture was stirred at 0 °C for 10 min, then iodine pellets were added until the solution assumed a persistent orange color. The ice bath was removed and the reaction was stirred at room temperature for 1 h. Afterwards, sodium thiosulfate satd. aq. sol. (1 ml) was added and the biphasic mixture was vigorously stirred for 30 min. The layers were separated and the organic one was washed with H$_2$O (1 ml) and brine (1 ml), dried

over Na$_2$SO4, filtered and concentrated. The crude was purified by flash column chromatography twice (first column EtOAc/hexane 1:8, R*f*: 0.38; second column CH$_2$Cl$_2$/hexane 1:1, R*f*: 0.27) to give **SH-LS-244** as a colorless oil (20 mg, 14%). $^1$H NMR (CDCl$_3$, 500 MHz, mixture of rotamers), δ: 7.66–7.62 (m, 4 H), 7.44–7.35 (m, 6 H), 7.13–7.10 (m, 2 H), 6.88–6.84 (m, 2 H), 4.13–4.11 (m, 0.4 H), 3.89 (br. s, 0.4 H), 3.77-3.75 (m, 1.8 H), 3.71–3.67 (m, 1 H), 3.64–3.62 (m, 1 H), 3.57–3.53 (m, 0.4 H), 3.42–3.40 (m, 1 H), 2.31–2.23 (m, 1 H), 1.95–1.92 (m, 2 H), 1.89–1.87 (m, 1 H), 1.49 (s, 3.6 H), 1.33 (s, 5.4 H), 1.31–1.26 (m, 10 H), 1.05 (s, 9 H), 0.88 (t, $J = 7.0$ Hz, 3 H) ppm. $^{13}$C NMR (CDCl$_3$, 125 MHz, mixture of rotamers), δ: 154.2, 143.0, 138.8, 137.5, 135.6, 133.7, 133.5, 133.4, 133.3, 129.7, 129.6, 128.8, 127.7, 127.3, 127.1, 125.9, 79.5, 79.2, 65.4, 65.2, 63.6, 62.2, 47.0, 46.3, 45.4, 38.3, 37.8, 32.8, 31.8, 31.7, 30.1, 29.7, 29.2, 29.1, 29.0, 28.9, 28.6, 28.4, 28.3, 26.9, 24.1, 22.6, 19.3, 14.1 ppm. HRMS (ESI) calcd. for C$_{40}$H$_{56}$N$_3$O$_3$Si (M + H)$^+$ 654.40855, found 654.40964.

**SH-LS-244**                                                                          **SH-LS-222**

### *tert*-butyl (2*S*,3 *R*)-3-(4-(3-heptyl-*3H*-diazirin-3-yl)phenyl)-2-(hydroxymethyl)pyrrolidine-1-carboxylate (SH-LS-222)

TBAF (29 µL, 1.0 M in THF, 0.029 mmol, 1.1 eq.) was added to a solution of **SH-LS-244** (17 mg, 0.026 mmol) in dry THF (1 ml) in a flask wrapped in aluminum foil. The reaction was then stirred at r.t. for 1 h. Afterwards, the mixture was diluted with NaHCO$_3$ satd. solution (0.5 ml) and extracted with EtOAc (3 × 0.5 ml). The organic layers were washed with brine (0.5 ml), dried over MgSO$_4$, filtered and concentrated. The crude was purified by flash column chromatography (EtOAc/hexane 1:4, R*f*: 0.14) to give **SH-LS-222** as a colorless oil (10 mg, 91%). α$^{25}$$_D$ -2.4 (*c* 0.5, CHCl$_3$). IR (neat), ν$_{max}$: 3409, 2926, 2857, 1667, 1516, 1455, 1401, 1365, 1252, 1166, 1116, 1083, 1054, 911, 847, 772, 733, 568 cm$^{-1}$. $^1$H NMR (CDCl$_3$, 500 MHz), δ: 7.19 (d, $J = 8.3$ Hz, 2 H), 6.90 (d, $J = 8.3$ Hz, 2 H), 4.95 (d, $J = 7.2$ Hz, 1 H), 3.91 (t, $J = 7.3$ Hz, 1 H), 3.77–3.68 (m, 2 H), 3.62–3.58 (m, 1 H), 3.37–3.31 (m, 1 H), 2.91–2.86 (m, 1 H), 2.13 (br. s, 1 H), 1.96–1.87 (m, 3 H), 1.49 (s, 9 H), 1.29–1.22 (m, 10 H), 0.87 (t, $J = 7.0$ Hz, 3 H) ppm. $^{13}$C NMR (CDCl$_3$, 125 MHz), δ: 156.8, 139.9, 138.2, 127.5, 126.1, 80.6, 66.9, 65.9, 47.4, 47.0, 32.9, 31.7, 30.1, 29.7, 29.2, 29.1, 29.0, 28.4, 24.1, 22.6, 14.0 ppm. HRMS (ESI) calcd. for C$_{24}$H$_{37}$N$_3$O$_3$Na (M+Na)$^+$ 438.2727, found 438.2729.

**SH-LS-222**                                                                          **SH-LS-249**

### *tert*-butyl (2*S*,3 *R*)-3-(4-(3-heptyl-*3H*-diazirin-3-yl)phenyl)-2-((prop-2-yn-1-yloxy)methyl)pyrrolidine-1-carboxylate (SH-LS-249)

NaH (2 mg, 60%, 0.048 mmol, 2 eq.) was added to a solution of **SH-LS-222** (10 mg, 0.024 mmol) in dry THF (0.5 ml) in a flask wrapped in aluminum foil at 0 °C. The resulting mixture was stirred

for 1 h at the same temperature, before adding propargyl bromide (9 μL, 80% sol. In toluene, 0.072 mmol, 3 eq.). Then, the reaction was stirred at r.t. overnight, before being quenched with $H_2O$ (0.25 ml). The product was extracted with EtOAc (2 ×1 ml) and the combined organic layers were dried over $MgSO_4$, filtered and concentrated. The crude was purified by flash column chromatography (EtOAc/hexane 1:8, R$f$: 0.31) to give **SH-LS-249** as a colorless oil (6 mg, 56%). IR (neat), $\nu_{max}$: 2927, 2856, 1692, 1455, 1392, 1366, 1255, 1168, 1110, 845, 771, cm$^{-1}$. $^1$H NMR (CDCl$_3$, 500 MHz, mixture of rotamers), δ: 7.16 (d, $J$ = 8.3 Hz, 2 H), 6.88 (d, $J$ = 8.3 Hz, 2 H), 4.17 (dd, $J$ = 15.9, 2.4 Hz, 1 H), 4.12 (dd, $J$ = 15.9, 2.4 Hz, 1 H), 3.95 (br. s, 0.4 H), 3.83 (br. s, 1 H), 3.69–3.58 (m, 2.6 H), 3.47 (dd, $J$ = 11.9, 6.9 Hz, 1 H), 3.37 (br. s, 1 H), 2.41 (br. s, 1 H), 2.28–2.22 (m, 1 H), 1.94–1.91 (m, 2 H), 1.89–1.86 (m, 1 H), 1.48 (s, 9 H), 1.32–1.21 (m, 10 H), 0.87 (t, $J$ = 7.0 Hz, 3 H) ppm. $^{13}$C NMR (CDCl$_3$, 125 MHz, mixture of rotamers), δ: 154.3, 137.6, 127.2, 125.9, 80.4, 79.8, 74.4, 73.2, 69.6, 68.8, 65.9, 63.3, 58.4, 46.4, 45.9, 45.4, 32.4, 31.7, 31.4, 30.1, 29.7, 29.2, 29.1, 29.0, 28.5, 24.1, 22.6, 14.1 ppm. HRMS (ESI) calcd. for $C_{27}H_{39}N_3O_3Na$ (M+Na)$^+$ 476.28836, found 476.28909.

**SH-LS-249** → HCl 4 N, dioxane → **SH-LS-253**

### (2S,3R)-3-(4-(3-heptyl-*3H*-diazirin-3-yl)phenyl)-2-((prop-2-yn-1-yloxy)methyl)pyrrolidine hydrochloride (SH-LS-253 or 893-PAL)

HCl (66 μL, 4 M in dioxane) was added to a solution of **SH-LS-249** (6 mg, 0.013 mmol) in dry dioxane (66 μL) in a flask wrapped in aluminum foil. The reaction was stirred at RT for 1 h. The solution was then concentrated *in vacuo* in several cycles co-distilling with dry dioxane. The crude was triturated three times in a diethyl ether/hexane mixture to give **SH-LS-253** as a pale yellow solid (4.6 mg, 88%). $^1$H NMR (CD$_3$OD, 500 MHz), δ: 7.37 (d, $J$ = 8.3 Hz, 2 H), 7.02 (d, $J$ = 8.3 Hz, 2 H), 4.29 (dd, $J$ = 16.0, 2.3 Hz, 1 H), 4.24 (dd, $J$ = 16.0, 2.3 Hz, 1 H), 3.77–3.75 (m, 1 H), 3.73 (dd, $J$ = 10.7, 2.8 Hz, 1 H), 3.66 (dd, $J$ = 10.7, 6.4 Hz, 1 H), 3.61–3.56 (m, 1 H), 3.45–3.35 (m, 2 H), 2.96 (t, $J$ = 2.3 Hz, 1 H), 2.48 (dtd, $J$ = 10.3, 7.5, 2.9 Hz, 1 H), 2.28–2.20 (m, 1 H), 1.99–1.96 (m, 2 H), 1.37-1.23 (m, 10 H), 0.92 (t, $J$ = 7.0 Hz, 3 H) ppm. $^{13}$C NMR (CD$_3$OD, 125 MHz), δ: 138.9, 137.2, 127.3, 126.1, 78.2, 75.7, 65.7, 64.7, 58.0, 44.9, 44.5, 32.1, 31.5, 29.6, 28.8, 28.6, 23.8, 22.2, 13.0 ppm. HRMS (ESI) calcd. for $C_{22}H_{32}N_3O$ (M)$^+$ 354.25399, found 354.25486.

**SH-LS-277** → 1) BnNH$_2$, ZnCl$_2$, MS 4 Å, toluene, reflux; 2) NH$_3$, HASA, MeOH; 3) I$_2$, Et$_3$N, CH$_2$Cl$_2$ → **SH-LS-280**

### 1-((2S,3R)-2-(((*tert*-butyldiphenylsilyl)oxy)methyl)-3-(4-(3-heptyl-*3H*-diazirin-3-yl)phenyl)pyrrolidin-1-yl)ethan-1-one (SH-LS-280)

Benzylamine (39 μL, 0.36 mmol, 3 eq.) and ZnCl$_2$ (1 mg, 0.007 mmol, 0.06 eq.) were added to a solution of **SH-LS-277** (69 mg, 0.12 mmol) in dry toluene (2 ml) containing 4 Å molecular sieves and the resulting mixture was refluxed for 8 h. Afterwards, the reaction was diluted with Et$_2$O (10 ml) and washed with H$_2$O (2.5 ml) and brine (2.5 ml). Then, the organic layer was dried over Na$_2$SO$_4$, filtered and concentrated *in vacuo*, resulting in a yellow oil. This intermediate imine was dissolved in dry MeOH (1.5 ml) and added dropwise to a flame-dried three-neck flask containing liquid ammonia (2 ml ca. condensed by connecting directly a tank to the flask and slowly adding gaseous ammonia), equipped with a cold finger at -78 °C and kept in a cool bath at the same temperature and under an argon atmosphere (balloon). The cool bath was removed and the resulting mixture was stirred for 2 h at ammonia refluxing point, by using the cold finger at -78 °C as condenser. Finally, a solution of hydroxylamine-*O*-sulfonic acid (22 mg, 0.19 mmol, 1.6 eq.) in dry MeOH (1 ml) was added dropwise and the reaction was allowed to slowly reach room temperature and stirred overnight. Afterwards, the mixture was concentrated, H$_2$O (2.5 ml) was added to the residue and the product was extracted with Et$_2$O (2 × 5 ml). The organic layers were washed with brine (2.5 ml), dried over MgSO$_4$, filtered and concentrated, resulting in a yellow oil. This intermediate diaziridine was dissolved in dry CH$_2$Cl$_2$ (1.5 ml) in a flask wrapped in aluminum foil and cooled down to 0 °C, before adding Et$_3$N (50 μl, 0.36 mmol, 3 eq.). The mixture was stirred at 0 °C for 10 min, then iodine pellets were added until the solution assumed a persistent orange color. The ice bath was removed and the reaction was stirred at room temperature for 1 h. Afterwards, sodium thiosulfate satd. aq. sol. (0.5 ml) was added and the biphasic mixture was vigorously stirred for 30 min. The layers were separated and the organic one was washed with H$_2$O (0.5 ml) and brine (0.5 ml), dried over Na$_2$SO4, filtered and concentrated. The crude was purified by flash column chromatography (EtOAc/hexane 1:1, R$f$: 0.30) to give **SH-LS-280** as a colorless oil (6 mg, 8%). $^1$H NMR (CDCl$_3$, 400 MHz, mixture of rotamers), δ: 7.64–7.60 (m, 4 H), 7.46–7.34 (m, 6 H), 7.10–7.06 (m, 2 H), 6.88 (d, $J$ = 8.4 Hz, 0.8 H), 6.85 (d, $J$ = 8.4 Hz, 1.2 H), 4.21–4.18 (m, 0.6 H), 4.04 (dd, $J$ = 10.3, 4.8 Hz, 0.6 H), 3.88 (dt, $J$ = 12.4, 7.9 Hz, 0.4 H), 3.83–3.79 (m, 0.4 H), 3.76 (dd, $J$ = 10.3, 2.5 Hz, 0.6 H), 3.69–3.57 (m, 2 H), 3.55–3.49 (m, 1 H), 3.44-3.37 (m, 0.4 H), 2.40–2.33 (m, 0.6 H), 2.31–2.25 (m, 0.4 H), 2.23–2.19 (m, 0.8 H), 2.06 (s, 1.8 H), 1.95–1.87 (m, 2.4 H), 1.84 (s, 1.2 H), 1.70-1.62 (m, 0.8 H), 1.30–1.26 (m, 8 H), 1.06 (s, 5.4 H), 1.05 (s, 3.6 H), 0.89-0.86 (m, 3 H) ppm.

**SH-LS-280** → TBAF, THF → **SH-LS-286**

### 1-((2S,3R)-3-(4-(3-heptyl-*3H*-diazirin-3-yl)phenyl)-2-(hydroxymethyl)pyrrolidin-1-yl)ethan-1-one (SH-LS-286 or 893-diazirine)

TBAF (11 μL, 1.0 M in THF, 0.011 mmol, 1.1 eq.) was added to a solution of **SH-LS-280** (6 mg, 0.010 mmol) in dry THF (0.5 ml) in a flask wrapped in aluminum foil. The reaction was then stirred at r.t.

for 30 min. Afterwards, the mixture was diluted with $NaHCO_3$ satd. solution (0.25 ml) and extracted with EtOAc (3 ×0.25 ml). The organic layers were washed with brine (0.25 ml), dried over $MgSO_4$, filtered and concentrated. The crude was purified by flash column chromatography (pure EtOAc, R*f*: 0.12) to give **SH-LS-286** as a colorless oil (3 mg, 80%). $^1H$ NMR ($CDCl_3$, 500 MHz), δ: 7.19 (d, $J = 8.4$ Hz, 2 H), 6.91 (d, $J = 8.4$ Hz, 2 H), 5.28 (dd, $J = 9.0, 1.8$ Hz, 1 H), 4.11 (t, $J = 7.4$ Hz, 1 H), 3.75-3.68 (m, 2 H), 3.61 (ddd, $J = 11.6, 7.4, 1.8$ Hz, 1 H), 3.55 (td, $J = 10.2, 6.4$ Hz, 1 H), 2.95–2.90 (m, 1 H), 2.24 (dtd, $J = 9.1, 6.4, 2.7$ Hz, 1 H), 2.16 (s, 3 H), 2.06–1.96 (m, 1 H), 1.94–1.91 (m, 2 H), 1.29–1.20 (m, 10 H), 0.87 (t, $J = 7.0$ Hz, 3 H) ppm. $^{13}C$ NMR ($CDCl_3$, 125 MHz), δ: 171.7, 139.4, 138.5, 127.4, 126.2, 67.9, 65.8, 48.5, 47.1, 33.1, 31.7, 30.1, 29.2, 29.1, 29.0, 24.1, 23.1, 22.6, 14.1 ppm. HRMS (ESI) calcd. for $C_{21}H_{32}N_3O_2$ $(M + H)^+$ 358.2489, found 358.2486.

## Data availability

The mass spectrometry proteomics data have been deposited to the ProteomeXchange Consortium via the PRIDE repository with the dataset identifier PXD036429. Project Webpage: https://www.ebi.ac.uk/pride/archive/projects/PXD036429.

The source data of this paper are collected in the following database record: biostudies:S-SCDT-10_1038-S44318-025-00490-5.

## Peer review information

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

## Acknowledgements

We thank the following UCI colleagues for advice and instrument access: Tom Poulos, Celia Goulding, Wenqi Wang, Felix Grun, Gina Lee, and Adeela Syed. This work was supported by NIH R01 GM089919 and R01 CA254360 (ALE), NIH R21 CA178230 (ALE and SH), RSG-11-111-01-CDD from the American Cancer Society (ALE), US Army CDMRP grant BC14027 (ALE), an Anti-Cancer Challenge grant from the Chao Family Comprehensive Cancer Center (ALE), a Proof of Product grant from UCI Applied Innovation (ALE), the Ono Pharma Foundation Breakthrough Science Initiative (G-1902-0039, ALE), an unrestricted gift from Kairos Ventures (ALE), and NIH R35GM148350 to PK. VJ, ANM, SYV, and BTF received support from NCI T32 CA009054. SJ received support from the National Research Foundation of Korea and the American Diabetes Association (2021R1A6A3A-14039681 and 11-23-PDF-03). The authors also wish to acknowledge the support of the Chao Family Comprehensive Cancer Center Optical Biology Center and Experimental Tissue Resource shared resources, supported by the National Cancer Institute of the National Institutes of Health under award number P30 CA062203. The content is solely the responsibility of the authors and does not necessarily represent the official views of the National Institutes of Health. SH and PT acknowledge financial support from NSERC Canada. The Institute for Research in Immunology and Cancer (IRIC) receives infrastructure support from IRICoR, the Canadian Foundation for Innovation, and the Fonds de Recherche du Québec - Santé (FRQS). IRIC proteomics facility is a Genomics Technology platform funded in part by Genome Canada and Genome Québec. Partial funding for publishing this article open access was provided by the University of California Libraries under a transformative open access agreement with the publisher. The synopsis illustration and Figs. 1B, 4A,E, and Appendix Fig. S2B were created in BioRender.

## Author contributions

**Vaishali Jayashankar**: Conceptualization; Formal analysis; Validation; Investigation; Visualization; Methodology; Writing—original draft. **Peter Kubiniok**: Conceptualization; Formal analysis; Validation; Investigation; Visualization; Methodology; Writing—original draft; Writing—review and editing. **Alison N McCracken**: Conceptualization; Formal analysis; Validation; Investigation; Visualization; Methodology; Writing—original draft; Writing—review and editing. **Rebeca G Gentry**: Formal analysis; Validation; Investigation; Visualization; Methodology. **Kazumi H Eckenstein**: Formal analysis; Validation; Investigation; Visualization; Methodology. **Lorenzo Sernissi**: Resources; Validation; Investigation; Visualization; Methodology; Writing—original draft. **Vito Vece**: Resources; Validation; Investigation; Visualization; Methodology; Writing—original draft. **Jean-Baptiste Garsi**: Resources; Validation; Investigation; Visualization; Methodology; Writing—original draft. **Sarah Y Valles**: Formal analysis; Validation; Investigation; Visualization; Methodology; Writing—review and editing. **Sunhee Jung**: Formal analysis; Validation; Investigation; Methodology; Writing—original draft. **Natalie C Hoffman**: Formal analysis; Validation; Investigation; Visualization; Methodology. **Arielle S Perrochon**: Validation; Investigation; Visualization; Methodology. **Elizabeth M Selwan**: Formal analysis; Validation; Investigation; Visualization; Methodology; Writing—original draft; Writing—review and editing. **Brendan T Finicle**: Formal analysis; Validation; Investigation; Visualization; Methodology; Writing—original draft. **Mary Pitman**: Formal analysis; Investigation; Visualization; Methodology; Writing—original draft. **DaWei Lin**: Resources; Methodology. **Éric Bonneil**: Formal analysis; Investigation. **Ruijuan Xu**: Resources. **Cungui Mao**: Resources; Writing—review and editing. **Peter Kaiser**: Resources; Supervision; Methodology; Writing—original draft; Writing—review and editing. **David A Fruman**: Supervision; Conflict of interest management. **David Mobley**: Formal analysis; Supervision; Validation; Investigation; Visualization; Methodology; Writing—original draft; Writing—review and editing. **Cholsoon Jang**: Formal analysis; Supervision; Validation; Investigation; Methodology; Writing—original draft; Writing—review and editing. **Stephen Hanessian**: Conceptualization; Resources; Formal analysis; Supervision; Funding acquisition; Validation; Visualization; Methodology; Writing—original draft; Project administration; Writing—review and editing. **Pierre Thibault**: Conceptualization; Formal analysis; Supervision; Funding acquisition; Validation; Visualization; Methodology; Writing—original draft; Project administration; Writing—review and editing. **Aimee L Edinger**: Conceptualization; Formal analysis; Supervision; Funding acquisition; Validation; Visualization; Methodology; Writing—original draft; Project administration; Writing—review and editing.

Source data underlying figure panels in this paper may have individual authorship assigned. Where available, figure panel/source data authorship is listed in the following database record: biostudies:S-SCDT-10_1038-S44318-025-00490-5.

## Disclosure and competing interests statement

ALE and SH are inventors on a patent covering SH-BC-893. Other authors declare no competing interests. ALE, PT, and SH are co-corresponding authors. Requests for biological materials should be directed to ALE.

