## [Peer Review File · The EMBO Journal]

Sphingosine simultaneously inhibits nuclear import and activates PP2A by binding importins and PPP2R1A

Vaishali Jayashankar, Peter Kubiniok, Alison McCracken, Rebeca Gentry, Kazumi Eckenstein, Lorenzo Sernissi, Vito Vece, Jean-Baptiste Garsi, Sarah Valles, Sunhee Jung, Natalie Hoffman, Arielle Perrochon, Elizabeth Selwan, Brendan Finicle, Mary Pitman, Da-Wei Lin, Eric Bonneil, Ruijuan Xu, Cungui Mao, Peter Kaiser, David Fruman, David Mobley, Cholsoon Jang, Stephen Hanessian, Pierre Thibault, and Aimee Edinger

Corresponding author(s): Aimee Edinger (aedinger@uci.edu) , Pierre Thibault (pierre.thibault@umontreal.ca), Stephen Hanessian (shanessi@uci.edu)

Review Timeline:

Submission Date:	18th Apr 25
Editorial Decision:	7th May 25
Revision Received:	18th May 25
Accepted:	27th May 25

Editor: William Teale

Transaction Report:

This manuscript was transferred to The EMBO JOURNAL following peer review at another journal.

Reviewer 1 – lipids/sphingolipids expert

In this manuscript the authors use structure activity based development of sphingosine derivatives which they attach to beads for protein purification or modify them for cross-linking studies. The development of the probes is well thought through and has likely been an important aspect of the success of this work. Affinity techniques have been used before, but never with such clear results, likely due to the improved probe development. Sphingosine and its derivatives have long been thought to have intracellular targets, but aside from the kinases where they are substrates, important targets have not been identified.

Furthermore, most studies on sphingosine and their analogues have never been clear about whether the sphingosine was the ligand or its phosphorylated or N-acylated form. Both modifications are clearly ruled out here since N-acylation reduces probe activity enormously and non-phosphorylated analogs are also active. Therefore, the conclusion that sphingosine and sphingosine analogs are the ligands is well documented.

We thank Reviewer 1 for highlighting the novelty, impact, and rigor of the work presented and appreciate Reviewer 1's statement that our work clearly discriminates the structure activity relationships among different physiologically relevant molecules related to sphingosine.

Furthermore, the use of chemiproteomics as an unbiased tools to identify the targets through binding in different cell types and in comparison to inactive analogs provides an unbiased approach that is quite convincing. One limitation of the technique is that it is likely that they have missed most transmembrane protein targets. Consistent with this they do not make claims that the target identification is exhaustive.

They focus on two classes of targets that they have found, regulatory subunits of PP2A and importins. PP2A genetic interactions with sphingoid bases has been shown previously for endocytosis in yeast, but the protein target was not shown and was suggested to be protein kinases. Substantial progress has been made with their in vitro work that shows that the regulatory subunit of PP2A, PPP2R1A, is unfolded in response to binding. Unfolding of the subunit should lead to increased PP2A activity and lower protein phosphorylation, which seems to be the case. We don't have a complete molecular detail of how sphingosine unfolds PP2R1A. It binds to it as it can be crosslinked to several residues, but how the unfolding occurs is not clear. It seems to me that they have used rather high (although not excessive) amounts of compounds for their cross-linking experiments and this could be one reason why they have found several residues. In order to identify the primary binding site, if there is one, I would suggest using lower amounts of cross-linkable compound and/or shorter incubation times. It seems that they decided not to concentrate on this mechanism, but rather the effects of unfolding on sphingosine activity in potential cancer therapeutics. At some point it would be worthwhile to understand how a small compound like sphingosine can lead to regulated protein unfolding.

We have revised Fig. 2 to include only amino acids that were crosslinked at 5 μ M 893-diazirine. Residues crosslinked at 20 μ M (which are generally not as well conserved) are still included in Supplementary Table 6. We agree that understanding how sphingosine-like molecules interact with Ppp2r1a and induce activation of the phosphatase is of great interest and this is the subject of on-going studies.

The second set of targets are the importins. This was a completely unexpected and exciting finding. They also used unbiased proteomics to identify which nuclear proteins are affected by treatment and have come up with some that might explain the effects on cell proliferation in cancer cells. They provide evidence that importin activity is inhibited by the compounds, probably through a reduction in association with nuclear pores. Some of the proteins affected in their import are JUN, MYC and YAP. They do quantitative immunofluorescence to examine the extent of nuclear localization with the inhibitors and show that they give the expected phenotype.

They also addressed the question of whether the PP2A and importin activities are additive or redundant in nuclear import. Quite interestingly, they seem to be additive which means that the effects of the compounds on nuclear import are two fold. Inhibiting PP2A is probably increasing phosphorylation of these transcription factors which leads to increased ubiquitination and degradation by the proteasome. Independent to this, inhibition of the importing directly reduces nuclear import.

Finally, they show that ceramide has different effects than sphingosine and thus works by a different mechanism. This seemed already evident to me from their studies earlier in the manuscript. More important is the demonstration that sphingosine levels can actually reach the required levels in cells, which they did with the help of mass spectrometry and inhibitors of sphingosine metabolism.

Reviewer 1 highlights that we showed in the original submission that the relevant phenotypes are detected at physiologically relevant levels of sphingosine. Additional information supporting this conclusion has been added to address comments from Reviewer 2 (Fig. 6 and ED Fig. 10a).

All in all, this is an excellent manuscript that has identified long-sought after intracellular targets of sphingoid bases and puts intracellular sphingoid base signaling on firm ground.

We thank Reviewer 1 for their thorough review.

Reviewer 2 – sphingolipids

In this study, Jayashankar et al report on interesting results on regulation of PP2A regulatory subunit (PPP2R1A) and other proteins by sphingosine and sphingosine like molecules. Clever results in Figure 4 that demonstrate the existence of more than one target for the SH compound. The results support this regulation in cells, however, the physiologic significance of these results is in serious doubt.

Major Points

1. A critical point is the issue of specificity of action of these molecules. The diversity of structures and variations suggest that these are non-specific effects of amphipathic molecules that contain an amino group. The structures of the less potent molecules seem to either destroy the hydrophobic preponderance of the active molecules and/or remove the charge of the amino groups. Therefore, the main commonality is the presence of a hydrophobic amine; a phenomenon noted decades ago for the non-specific actions of sphingosine and other hydrophobic amines. As such, one would predict that stearylamine would work just as well.

As highlighted by Reviewer 1, we have now shown that sphingolipids with biologically relevant structural differences (sphingosine and ceramide, which do and do not have a positive charge respectively) do not bind the same targets. Establishing this structure-function relationship provides new insights into the action of biologically active sphingolipids.

In addition to the original studies with ceramide, we now include ED Fig. 10b,c showing that:

- 1) The structurally similar precursor for sphingosine, 3-keto-sphinganine, does not efficiently bind these targets despite its structural similarity to sphingosine.*
- 2) Shortening (C12) or lengthening (C20) the hydrocarbon chain reduces target binding relative to natural sphingosine (C18). In addition, altering the position of the double bond in sphingosine (d18:1 4E to 14Z) also altered target binding. These results indicate there is specificity to binding within the hydrophobic region.*

As mentioned above, the manuscript also includes data showing that the sphingosine metabolite ceramide does not bind these targets. It is biologically significant that N-acylation (sphingosine's conversion to ceramide) disrupts binding (Fig. 5a,b and ED Fig. 9f-j). Our finding that these two tumor suppressive sphingolipids, sphingosine and ceramide, engage different targets provides evidence that had been lacking that these tumor suppressor sphingolipids have distinct modes of action. Our published finding that more nuclear proteins were dephosphorylated in response to ceramide (no importin engagement) than 893 (importin inhibition) is consistent with the work reported here. From Kubiniok et al. (1), classification of proteins whose phosphorylation state was affected by these molecules:

D

Because Reviewer 2 raised the question, we tested whether stearylamine causes similar conformational changes to 893. Stearylamine does not phenocopy sphingosine or SH-BC-893; the first derivative plots do not shift.

We do not include data with stearylamine in the manuscript because inclusion of this compound might be confusing to readers as stearylamine is neither physiologically nor pharmacologically relevant. In sum, there is strong evidence that the sphingolipids studied here demonstrate biologically relevant specificity in target binding.

Since most/all high affinity lipid-protein interactions show significant specificity (even stereospecificity), some indication of stereospecificity of action of sphingosine (which exists in 4 stereoisomers only one of which is natural) would go a long way in suggesting a specific lipid-protein interaction.

While stereospecificity is often observed, it was not observed for these sphingosine like compounds (please see (2,3)). As Reviewer 2 points out, only the D-erythro-sphingosine stereoisomer is abundant in cells; it is possible that there was little evolutionary pressure to maintain stereospecificity.

Therefore, the data presented overall suggest a non-specific protein lipid interaction. In this regard, the bead-binding studies need a legitimate control for specificity (i.e. a very closely related molecule that does not bind and is not active in cells).

We respectfully point out that the negative controls in the chemoproteomics (bead binding) study included six molecules that meet the reviewer's criteria: "very closely related molecules that do not bind and are not active in cells" (structures and activity data are in ED Fig. 1a-f of the original and revised submission). We included a negative control compound where the hydrophobic region was disrupted, 893-ketone, and several where hydrophobic non-specific interactions would still occur. Reviewers 1 and 3 specifically commented on the rigor and appropriateness of our orthogonal chemoproteomics approach using 4 active analogs paired with 6 inactive analogs. Notably, many of the negative control compounds mimic differences between sphingosine and ceramide, and additional specificity controls were added in response to Reviewer 2's concerns (ED Fig. 10b,c).

Moreover, the structural modification at the 1-OH destroys a key feature of sphingosine.

As discussed in the introduction, we intentionally disrupted phosphorylation to avoid the bradycardia and lymphocyte sequestration associated with FTY720 (3); these actions would not be desirable in an anti-cancer agent. This structural modification, as pointed out by Reviewer 1, also helps us reveal the functions of sphingosine itself rather than those of S1P, a key goal of this project. Very little is known about the specific actions of sphingosine as most work has focused on S1P and ceramide.

Our work advances our understanding of sphingolipid biology by revealing novel functions of the abundant sphingolipid sphingosine, differentiating the effects of sphingosine and ceramide, and by providing insight into the established anti-cancer (4) and anti-obesity (5) actions of the sphingosine mimic 893.

2. Based on the above, the physiologic significance of these interactions is not demonstrated. The concentrations used (1-20 μM) are significantly higher than the measured cellular concentrations of sphingosine (in the order of low nM). The authors' own results negate the assertion that the concentrations of exogenous sphingosine are within the physiologic range.

We respectfully point out that Reviewer 2's statement is not correct: in the literature and in our study, cellular sphingosine concentrations are micromolar, not nanomolar. This confusion likely stems from the fact that the cellular concentration of sphingolipids is rarely reported with volume as the denominator.

The Hannun and Merrill groups are acknowledged experts in the measurement of sphingolipid levels. Older papers from these groups (6,7) explicitly state that the cellular concentrations of sphingosine in cells are micromolar. In most papers however, sphingosine levels are normalized to cellular protein levels. Some examples:

- In (8) from the Hannun/Obeid labs, sphingosine levels in MCF-7 human breast cancer cells are reported to be 3 pmol/mg protein and increase to 7 pmol/mg protein with PMA treatment.
- In (9), tumor sphingosine levels were reported at 25 pmol/mg protein, while kidney levels were 155 pmol/mg protein.
- In (10), human Molt-4 leukemia cells have basal sphingosine levels of 15 pmol/mg protein and this increases to 35 pmol/mg with actinomycin D-induced SK1 degradation.
- In another paper from the Hannun group (11), basal sphingosine levels in HCT116 human colorectal cancer cells were 50 pmol/mg protein.

The sphingosine levels reported in these papers, 3-155 pmol/mg protein, are consistent with our measurements of 12-113 fmol/ μg protein (equivalent to 12-113 pmol/mg protein). We expressed sphingosine levels as fmol/ μg rather than pmol/mg to simplify the conversion to molarity as cellular volumes are routinely reported in fL. The volume of cells is 2,000-3,000 fL (<http://book.bionumbers.org/how-big-is-a-human-cell/>); the volume of a HeLa cell has been estimated at 2,600 fL (12) which is in line with our measurement (ED Fig. 10a). Reference (13) suggests there is 500 pg protein/cell, while ThermoFisher reports 300 pg of protein per Hela cell also consistent with our findings in ED Fig. 10a. Bionumbers lists similar values (<https://bionumbers.hms.harvard.edu/bionumber.aspx?id=113240&ver=3>). **Using these measurements to calculate molarity (see ED Fig. 10a), we find basal sphingosine levels in the cells we use are 4-16 μM . The 3-155 pmol/mg values reported in the literature would translate to around 1-40 μM and are therefore consistent with our results.**

In (14) from the Spiegel/Millstein groups, SK1 WT and KO MEFs were evaluated for basal sphingosine levels. They report 250-400 pmol/mg sphingosine which translates to 65-105 μM prior to the addition of exogenous sphingosine. Our values are slightly lower, basally 13-15 μM , which may reflect differences in serum composition or culture conditions.

The Merrill lab reports similar values for tissues. In (15), sphingosine in the rat liver was reported as 7 nmol/g tissue while brain levels were 5 nmol/g tissue. Assuming a gram of tissue = 1 ml, this is also consistent with μM levels of sphingosine.

$$\frac{7 \text{ nmol}}{\text{g}} \times \frac{1 \text{ g}}{1 \text{ ml}} \times \frac{1000 \text{ ml}}{1 \text{ L}} \times \frac{1 \mu\text{mol}}{1000 \text{ nmol}} = 7 \mu\text{M}$$

In another study from the Merrill group (16), liver sphingosine was measured to be 15 nmol/g or 15 μM .

Molarity calculations were included in the text of the original submission but may have been overlooked during review. To increase clarity, we now show the relevant math in ED Fig. 10a and include a table that summarizes the data needed to make this calculation. We hope that the new presentation makes the physiological relevance of our results easier to see. We have also added the above citations and a clearer statement that sphingosine concentrations in cells are micromolar.

In fact, the results in Fig 5 d&e show a vast increase in endogenous sphingosine after 10uM (comparing 5d with 6g, there seems to be a nearly 1000 fold increase). Actually, the levels in the WT cells are still several fold higher than basal at the later time points, yet these do not exert any of the effects they want to attribute to sphingosine.

The criticism is that although sphingosine is elevated in SK1 WT MEFs at 6 h compared to baseline, the expected sphingosine-induced phenotypes are mild. SK1 WT/KO MEFs have been used by many groups, and it is not uncommon that the effects of sphingosine are lost at 6 h. For example, in the figure at right from (14), WT MEFs treated with 10 μ M sphingosine as in our study lost the phenotype of interest at 6 h similar to what we observed for nuclear YAP (Fig. 5g,h). This group did not report sphingosine levels over time, only basally, so the kinetics of sphingosine decline were not clear.

We analyzed nuclear YAP levels at 6 h because time is required for nuclear export to decrease nuclear YAP once import is inhibited (the faucet is off, but the sink must drain). In Fig 5k-m where nuclear p65 was measured in SK1 WT and KO MEFs 30 min after TNF α treatment (sink starts out empty, faucet is turned on), we were able to observe a more robust effect of sphingosine in WT MEFs because the timepoint was short (high sphingosine levels, little time for negative feedback).

As the reviewer states, adding 10 μ M exogenous sphingosine dramatically elevates cellular sphingosine. However, we have added new experiments where no exogenous sphingosine is supplied that produced similar results:

- 1) Chemical SK1 inhibition (100 nM PF-543) in Fig 6c-f induces a 3-fold change in sphingosine that causes a statistically significant reduction in nuclear NF-kB.
- 2) Inducing ACER1 expression in HeLa cells (17) (Fig. 6g-j) increases sphingosine levels about 5-fold and reduces nuclear JUN and YAP.

These experiments would also elevate sphingosine in physiologically relevant cellular compartments.

To summarize, contrary to Reviewer 2's initial conclusion, we are working within the physiologically relevant range and have now demonstrated that endogenous sphingosine produces the expected phenotypes.

3. Moreover, while these compounds do regulate the levels/location of several proteins, there is little (or no) evidence that any of the direct targets actually mediates any specific function of these molecules. For example, what does PP2A mediate? Importin?

We appreciate the reviewer's comment and the opportunity to clarify our findings. The phenotypes described in Figures 3–6 are linked to PP2A and/or the karyopherin importins, specifically KPNB1, TNPO1, IPO5, and IPO7, by rigorous studies published by other groups. To address the reviewer's concern about specific functional roles, we have strengthened the manuscript in the following ways:

1. **Direct Evidence for Functional Roles:** Figure 4a, retained from the original submission, provides data demonstrating how PP2A and KPNB1 specifically affect nuclear proteins such as MYC and the AR, which are well-known drivers of prostate cancer progression. The data establish a direct link between the activity of these targets and key oncogenic pathways.

2. **Expanded Context in Extended Data Figure 7b:** We have added a new table (ED Fig. 7b) that highlights specific nuclear proteins affected by these targets. This table includes details on how sphingosine-like compounds modulate nuclear levels of proteins reliant on KPNB1, TNPO1, IPO5, and IPO7 for import. Each of the targets in this table are evaluated in the manuscript with the predicted effect being observed.
3. **Unbiased Nuclear Proteomics Data:** Unbiased nuclear proteomics revealed that treatment with SH-BC-893 reduces nuclear levels of a subset (18%) of the nuclear proteome with 75% of the affected proteins exhibiting reduced nuclear levels in the presence of SH-BC-893. These results are consistent with a selective decrease in nuclear import activity, supported by data showing the reduction in nuclear levels of ribosomal proteins and transcription factors (e.g., JUN, YAP) that depend on these specific importins. Supplementary Table 8 added to this revision specifically documents changes in ribosomal proteins that were shown previously to be transported by the four importins targeted by these compounds.
4. **Functional Redundancy Among Importins:** The co-regulation of nuclear proteins is likely attributable to functional redundancy among KPNB1, TNPO1, IPO5, and IPO7. Published studies confirm that these importins are collectively responsible for transporting ribosomal proteins and transcription factors such as JUN into the nucleus. Thus, published work from groups studying nuclear import has established that these particular importins are functionally related providing a potential explanation for their co-regulation by sphingosine.
5. **Validation through Functional Assays:** Additional experiments highlight the dual impact of sphingosine-like compounds on PP2A activation and importin inhibition. This dual action not only confirms target engagement but also establishes how these targets collectively mediate reductions in nuclear MYC, JUN, and AR levels, further inhibiting prostate cancer drivers. Additional experiments where nuclear import of the well validated Kpnb1 target NF- κ B is acutely triggered by TNF provide a direct measure of import that is not influenced by export rates.

We have added additional citations of these relevant publications throughout the manuscript to provide a robust framework linking these observations to established biological pathways. These revisions, together with the new data and supplementary materials, aim to address the reviewer's concern and clarify the direct functional roles of PP2A and importins in mediating the observed phenotypes.

4. Any genetic evidence to support a role for cellular sphingosine in performing these functions (even simply overexpressing one of the ceramidases)?

We appreciate this suggestion. New data (Fig. 6g-j) from cells that inducibly express ACER1 (alkaline ceramidase 1, produces sphingosine from ceramide) strengthens our conclusions and is consistent with our experiments using SK1 WT/KO MEFs (Fig. 4) and PF-543 (Fig. 6a-f). Inducing ACER1 expression elevates sphingosine and reduced nuclear levels of both YAP and JUN (Fig. 6g-j) consistent with our other studies.

5. It is not clear if there was any binding to the HEAT domains (Fig. 2) The authors do not develop this hypothesis of interaction.

We have added a column to Supplementary Table 6 showing which HEAT repeat includes the crosslinked amino acids shown in Fig. 2. All crosslinked amino acids are found in HEAT repeats with the exception of D5 in Ppp2r1a. We agree that understanding how sphingosine-like molecules interact with these targets is of great interest and this is the subject of on-going studies.

6. Competition binding with regular compounds (no alkyne) has opposite results. The authors conclude it is due to a cooperative binding. This should be easy to validate.

Isothermal calorimetry (ITC) can be used to measure cooperativity. However, our results with ITC were uninterpretable. This makes sense because ligand binding triggers protein unfolding which causes major artifacts in ITC. We decided against attempting biolayer interferometry (Octet Red instrument) for the same reason – unfolding complicates the interpretation of results. While we elected to include the surprising competition data (ED Fig. 4g,h) to raise the interesting possibility of cooperative binding, this data is tangential to the main conclusions.

7. The results in Figure 3A could be easily interpreted as damage to membranes and leakiness to proteins. In this context, where did Jun (and YAP) go after using SH-BC-893?

Nuclear leakiness is not consistent with the results reported in Fig. 3a,b. It is unclear how the two separate lipid bilayers that make up the nuclear envelope could become leaky to only 18% of non-histone nuclear proteins (and the same 18% in 3 biological replicates). If proteins were leaving the nucleus non-specifically (leaking across both bilayers), proteins would be lost from the nucleus at random. We have also demonstrated in a published study that SH-BC-893 does not permeabilize endosomes under our experimental conditions (18). Our unpublished data also indicates that 893 does not damage membranes. Galectin-9 is recruited to sites of membrane damage (UNC1017938A is an endosomal escape agent (19), LLOMe (20) is used to damage membranes) membrane damage is not observed in 893-treated cells at 6 h.

To answer where JUN and YAP go, YAP can be seen to accumulate in the cytosol of 893-treated mPCE cells (Fig. 3b). For JUN, Western blotting suggests that total cellular JUN levels are not changing (ED Fig. 7c). Proteins dispersed in the cytosol can be more difficult to detect by immunofluorescence microscopy. Other proteins that display similar behavior (not detectable until they are recruited to and concentrated on membranes) include LC3 and Drp1.

8. Fig. 6. It is unfortunate that 10uM PF-543 was used as this inhibitor inhibits SK1 in low nM. A concentration of 10-100nM would be much more convincing than 10uM. In this context, are the effects of PF actually mediated by sphingosine (try to simultaneously disrupt or deplete levels of sphingosine). Also, why didn't the authors evaluate the levels of nuclear YAP where the responses seem a bit higher (but still with a lot of variation).

We have repeated this experiment using 100 nM PF-543 (Fig. 6c-f). Sphingosine is elevated and TNFa-induced NFkB translocation is inhibited as expected.

The box and whisker presentation makes it clear that nuclear protein levels are heterogeneous for every protein we examined. The p65 signal is robust and translocation in response to TNFa produces a large fold-change that makes it easier to distinguish partial inhibition by endogenous sphingosine (a similar approach was used in Fig. 5k-m for this reason - in a short experiment, the effects of sphingosine are apparent in SK1 WT cells).

***In conclusion,** we appreciate the reviewer generously offering their valuable time to review the manuscript and provide feedback. We have improved the manuscript by adding new data and improving the presentation of the data to address the concerns of Reviewer 2.*

Reviewer 3 – nuclear transport/importins

Jayashankar et al identified the targets (PPP2R1A, KPNB1, TNPO1, IPO5 and IPO7) of sphingosine-like compounds (SH-BC-893, SH-LS-200, FTY720, and phytosphingosine) through proteomics with cell lysates and photoaffinity labeling in situ. They found that the compounds decreased the Tm's of recombinant PPP2R1A and KPNB1, decreased the proteins' helical content and seem to cause their partial unfolding. In UV-crosslinking analysis, competition with unmodified compounds increased crosslinks, suggesting cooperative binding or that initial binding exposed new binding sites; consistent with the compounds causing

partial unfolding of their targets. Partial unfolding of PPP2R1A may re-position the B and C subunits to promote catalysis and/or substrate-binding. Partial unfolding/conformational change of KPNB1 may disrupt its interactions with nucleoporins and/or importin- α . Regardless of the exact modes of partial unfolding, the authors show that the sphingosine-like compounds activate PP2A and inhibit nuclear import functions of KPNB1, TNPO1, IPO5 and IPO7.

The manuscript is very well-written and the results are presented in a clear manner. Most importantly, the results are convincing and the conclusions appropriate. The findings the sphingosine-like compounds studied bind and partially unfold PPP2R1A and KPNB1, and likely TNPO1, IPO5 and IPO7 that are related to KPNB1. **The work is exciting and suitable for publication in Nature Cell Biology.**

Additional comments:

- 1) It would be useful if the authors could comment on whether other members of the karyopherin-beta family of importins, biportins and exportins were also detected in their proteomics analysis. If not, it'd be good if they could discuss why only KPNB1, TNPO1, IPO5 and IPO7 are targets.

Extended Data Figure 1g now highlights both the abundance of each karyopherin family member in the proteome and the fold enrichment by each of the chemoproteomics ligands for both FL5.12 cells (hematopoietic) and mPCE (prostate cancer).

*We hypothesize that the **functional redundancy of KPNB1, TNPO1, IPO5 and IPO7** explains their co-regulation by sphingosine-like compounds. Interestingly, JUN and ribosomal proteins were previously shown to rely on these 4 specific importins for nuclear import (21,22). These cargos offer regulatory points for homeostatic growth control. JUN promotes oncogenesis and growth (23–28). Control of ribosomal assembly would also regulate energy use and growth. To make these functional links clearer, we have added ED Fig. 7b and Supplemental Table 8 showing that the nuclear levels of many ribosomal proteins were reduced (nuclear proteomics), and many ribosomal proteins were isolated by the PAL probe along with KPNB1, TNPO1, IPO5, and IPO7 (consistent with ribosomal proteins being cargo).*

We detected another functional group of nuclear proteins that we speculate might be coordinately regulated by this group of importins: metabolic enzymes with “moonlighting” roles in the nucleus providing “ink” for epigenetic writers (29,30). Supplemental Table 9 highlights that many metabolic enzymes are lost from the nucleus and/or present in the PAL dataset (potential cargos for these importins) and thus might be regulated as group by sphingosine-like compound 893.

- 2) The authors connect PPP2R1A to the four importins as structurally homologous, but the importins are significantly more homologous to other importins, biportins and exportins, so the structural homology of PPP2R1A to the importins as reason for binding sphingosine-like compounds doesn't hold up. There are thousands of HEAT repeat proteins. What other characteristics are common between the five HEAT repeat proteins that they authors have identified as targets, other than they are comprised of HEAT repeats?

At this time, we are unsure why only these HEAT repeat proteins bind sphingosine-like compounds. We anticipate that further mapping the binding site in future studies will help us address this important question.

- 3) The authors mentioned that Importazole also decrease the Tm of KPNB1. It would be useful if they could describe what is known about how Importazole inhibit KPNB1, and compare with what they have learned about the mechanism of the sphingosine-like compounds acting on KPNB1.

Thank you for this suggestion, we have revised the text to include this information. Importazole (IPZ) was isolated in a “FRET-based, high-throughput small molecule screen for compounds that interfere with the interaction between RanGTP and importin β ”(31). Somewhat confusingly, IPZ reduced FRET between CFP-Ran and YFP-Kpnb1 but did NOT reduce RanGTP-Kpnb1 binding in pulldown assays. Because IPZ reduced

the Tm of Kpnb1 in a thermal shift assay, the authors concluded that “importazole binding [to Kpnb1] causes a conformational change that disrupts the CFP-RanGTP/YFP-importin β FRET interaction without preventing binding.” In our study, the 893-PAL probe pulled down Ran, but not Kpna2, from intact cells (Supplementary Table 2). This result could be consistent with a mechanism of action similar to importazole, but determining precisely how 893 and related compounds inhibit importins will require additional studies.

We thank Reviewer 3 for their time and helpful comments.

REFERENCES FOR RESPONSE TO REVIEW

1. Kubiniok P, Finicle BT, Piffaretti F, McCracken AN, Perryman M, Hanessian S, et al. Dynamic Phosphoproteomics Uncovers Signaling Pathways Modulated by Anti-oncogenic Sphingolipid Analogs. *Mol Cell Proteomics*. 2019 Mar;18(3):408–422.
2. Perryman MS, Tessier J, Wiher T, O’Donoghue H, McCracken AN, Kim SM, et al. Effects of stereochemistry, saturation, and hydrocarbon chain length on the ability of synthetic constrained azacyclic sphingolipids to trigger nutrient transporter down-regulation, vacuolation, and cell death. *Bioorg Med Chem*. 2016 Sep 15;24(18):4390–4397.
3. Chen B, Roy SG, McMonigle RJ, Keebaugh A, McCracken AN, Selwan E, et al. Azacyclic FTY720 analogues that limit nutrient transporter expression but lack S1P receptor activity and negative chronotropic effects offer a novel and effective strategy to kill cancer cells in vivo. *ACS Chem Biol*. 2016 Feb 19;11(2):409–414.
4. Kim SM, Roy SG, Chen B, Nguyen TM, McMonigle RJ, McCracken AN, et al. Targeting cancer metabolism by simultaneously disrupting parallel nutrient access pathways. *J Clin Invest*. 2016 Nov 1;126(11):4088–4102.
5. Jayashankar V, Selwan E, Hancock SE, Verlande A, Goodson MO, Eckenstein KH, et al. Drug-like sphingolipid SH-BC-893 opposes ceramide-induced mitochondrial fission and corrects diet-induced obesity. *EMBO Mol Med*. 2021 Aug 9;13(8):e13086.
6. Merrill AH. Cell regulation by sphingosine and more complex sphingolipids. *J Bioenerg Biomembr*. 1991 Feb;23(1):83–104.
7. Chao R, Khan W, Hannun YA. Retinoblastoma protein dephosphorylation induced by D-erythro-sphingosine. *J Biol Chem*. 1992 Nov 25;267(33):23459–23462.
8. Becker KP, Kitatani K, Idkowiak-Baldys J, Bielawski J, Hannun YA. Selective inhibition of juxtannuclear translocation of protein kinase C β 1 by a negative feedback mechanism involving ceramide formed from the salvage pathway. *J Biol Chem*. 2005 Jan 28;280(4):2606–2612.
9. Bai A, Liu X, Bielawski J, Hannun YA. Bioactive sphingolipid profile in a xenograft mouse model of head and neck squamous cell carcinoma. *PLoS One*. 2019 Apr 19;14(4):e0215770.
10. Taha TA, Osta W, Kozhaya L, Bielawski J, Johnson KR, Gillanders WE, et al. Down-regulation of sphingosine kinase-1 by DNA damage: dependence on proteases and p53. *J Biol Chem*. 2004 May 7;279(19):20546–20554.
11. Xu R, Wang K, Mileva I, Hannun YA, Obeid LM, Mao C. Alkaline ceramidase 2 and its bioactive product sphingosine are novel regulators of the DNA damage response. *Oncotarget*. 2016 Apr 5;7(14):18440–18457.
12. Zhao L, Kroenke CD, Song J, Piwnicka-Worms D, Ackerman JJH, Neil JJ. Intracellular water-specific MR of microbead-adherent cells: the HeLa cell intracellular water exchange lifetime. *NMR Biomed*. 2008 Feb;21(2):159–164.
13. Levy E, Slavov N. Single cell protein analysis for systems biology. *Essays Biochem*. 2018 Oct 26;62(4):595–605.
14. Lima S, Milstien S, Spiegel S. Sphingosine and sphingosine kinase 1 involvement in endocytic membrane trafficking. *J Biol Chem*. 2017 Feb 24;292(8):3074–3088.
15. Merrill AH, Wang E, Mullins RE, Jamison WC, Nimkar S, Liotta DC. Quantitation of free sphingosine in liver by high-performance liquid chromatography. *Anal Biochem*. 1988 Jun;171(2):373–381.

16. Zitomer NC, Mitchell T, Voss KA, Bondy GS, Pruett ST, Garnier-Amblard EC, et al. Ceramide synthase inhibition by fumonisin B1 causes accumulation of 1-deoxysphinganine: a novel category of bioactive 1-deoxysphingoid bases and 1-deoxydihydroceramides biosynthesized by mammalian cell lines and animals. *J Biol Chem*. 2009 Feb 20;284(8):4786–4795.
17. Lin C-L, Xu R, Yi JK, Li F, Chen J, Jones EC, et al. Alkaline Ceramidase 1 Protects Mice from Premature Hair Loss by Maintaining the Homeostasis of Hair Follicle Stem Cells. *Stem Cell Rep*. 2017 Nov 14;9(5):1488–1500.
18. Finicle BT, Eckenstein KH, Revenko AS, Anderson BA, Wan WB, McCracken AN, et al. Simultaneous inhibition of endocytic recycling and lysosomal fusion sensitizes cells and tissues to oligonucleotide therapeutics. *Nucleic Acids Res*. 2023 Feb 28;51(4):1583–1599.
19. Wang L, Ariyaratna Y, Ming X, Yang B, James LI, Kreda SM, et al. A Novel Family of Small Molecules that Enhance the Intracellular Delivery and Pharmacological Effectiveness of Antisense and Splice Switching Oligonucleotides. *ACS Chem Biol*. 2017 Aug 18;12(8):1999–2007.
20. Skowyra ML, Schlesinger PH, Naismith TV, Hanson PI. Triggered recruitment of ESCRT machinery promotes endolysosomal repair. *Science*. 2018 Apr 6;360(6384).
21. Waldmann I, Wälde S, Kehlenbach RH. Nuclear import of c-Jun is mediated by multiple transport receptors. *J Biol Chem*. 2007 Sep 21;282(38):27685–27692.
22. Jäkel S, Görlich D. Importin beta, transportin, RanBP5 and RanBP7 mediate nuclear import of ribosomal proteins in mammalian cells. *EMBO J*. 1998 Aug 3;17(15):4491–4502.
23. Edwards J, Krishna NS, Mukherjee R, Bartlett JMS. The role of c-Jun and c-Fos expression in androgen-independent prostate cancer. *J Pathol*. 2004 Oct;204(2):153–158.
24. Thakur N, Gudey SK, Marcusson A, Fu JY, Bergh A, Heldin C-H, et al. TGF β -induced invasion of prostate cancer cells is promoted by c-Jun-dependent transcriptional activation of Snail1. *Cell Cycle*. 2014;13(15):2400–2414.
25. Lukey MJ, Greene KS, Erickson JW, Wilson KF, Cerione RA. The oncogenic transcription factor c-Jun regulates glutaminase expression and sensitizes cells to glutaminase-targeted therapy. *Nat Commun*. 2016 Apr 18;7:11321.
26. Song D, Lian Y, Zhang L. The potential of activator protein 1 (AP-1) in cancer targeted therapy. *Front Immunol*. 2023 Jul 6;14:1224892.
27. Park J, Eisenbarth D, Choi W, Kim H, Choi C, Lee D, et al. YAP and AP-1 Cooperate to Initiate Pancreatic Cancer Development from Ductal Cells in Mice. *Cancer Res*. 2020 Nov 1;80(21):4768–4779.
28. Thomsen MK, Bakiri L, Hasenfuss SC, Wu H, Morente M, Wagner EF. Loss of JUNB/AP-1 promotes invasive prostate cancer. *Cell Death Differ*. 2015 Apr;22(4):574–582.
29. Boukouris AE, Zervopoulos SD, Michelakis ED. Metabolic enzymes moonlighting in the nucleus: metabolic regulation of gene transcription. *Trends Biochem Sci*. 2016 Jun 23;41(8):712–730.
30. Pan C, Li B, Simon MC. Moonlighting functions of metabolic enzymes and metabolites in cancer. *Mol Cell*. 2021 Sep 16;81(18):3760–3774.
31. Soderholm JF, Bird SL, Kalab P, Sampathkumar Y, Hasegawa K, Uehara-Bingen M, et al. Importazole, a small molecule inhibitor of the transport receptor importin- β . *ACS Chem Biol*. 2011 Jul 15;6(7):700–708.

Dear Aimee,

We have now received an arbitrator's report for your manuscript, which I have included below. As you will see, the opinion has been returned that the revised version of your manuscript satisfactorily and reasonably addresses the referees' concerns. Before I can finally accept the manuscript, there are some editorial points which need to be addressed. In this regard would you please:

- submit your manuscript as a Word file with the introduction section labeled as indicated below,
- include up to five keywords,
- include a 'Data Availability' section for the appropriate paragraph currently headed 'Author Information' (which should be removed); the relevant author information here should be included in a separate 'Disclosure and Competing Interests' section,
- remove the 'Author contributions' section from the manuscript,
- provide a URL for dataset PXD036429 in the data availability section,
- use an alphabetical (not numerical) reference format, using 'et al.' after 10 author names; remove DOIs, replacing with a consistent reference format,
- complete an author checklist; this can be downloaded in the Author Guidelines section of our website,
- include a Reagents and Tools table,
- remove funders and grants from the comments box of our online submission system and provide as separate entries via the 'More Funders' option,
- provide main figures as separate production-quality PDFs,
- check all figure panels are referred to in the main text consecutively; currently panels of Fig 2 are mentioned before all the panels of Fig 1,
- refer to Tables 1-7 as datasets, updating their nomenclature in all places (Dataset EV1-EV7: source file names, titles in the online submission system, main manuscript, ...), remove their legends from the manuscript file and provide in each Excel file as a separate sheet/tab; rename Suppl. Table 8-9 as Table EV1-EV2 in all places, also remove their legends from the main manuscript as they are already provided in each file,
- remove the 'Supplementary Information' section from the main manuscript,
- alter the nomenclature of the appendix file containing supplementary figures as 'Extended Data Fig. 1' etc.; provide a title page with Appendix title and a short ToC with page numbers; the correct nomenclature and callouts should be Appendix Figure S1-S10,
- provide exact p values in the legends of figures 1F, H; 3C, E, F, G; 4C, D, G, J; 5A, B, E, F, H, J, L, M; 6C, D, F, G, I, J; and extended data figures 3A, B, E, J, K, L, M, N; 7C-E; 8C, E, F, G; 9E, J; 10 C,
- indicate the statistical test used for data analysis in the legend of figure 3A,
- define box plots in terms of minima, maxima, percentile in the legends of figures 3C, E, F, G; 4C, D, G, J; 5H, J, L, M; 6F, I, J; and extended data figures 7D, E; 8E, 9E, J,
- define the nature of 'n' in the legends of figures 3A, extended data figures 3A, B; 10C
- define the error bars in the legend of extended data figure 10C,
- define the scale bar in extended data figure 1A, and
- include a scale bar and its definition for extended data figure 1B

The manuscript sections should be in the following order: Title page - Abstract & Keywords - Introduction - Results - Discussion - Methods - Data Availability - Acknowledgments - Disclosure Statement & Competing Interests - References - Figure Legends - (Main Tables with legends if applicable) - Expanded View Figure Legends.

We include a synopsis of the paper (see <http://emboj.embopress.org/>). Please provide me with a general summary image, a two sentence statement and 3-5 bullet points that capture the key findings of the paper.

I am looking forward to receiving your revised manuscript.

EMBO Press is an editorially independent publishing platform for the development of EMBO scientific publications.

Best wishes,

William

William Teale, PhD
Editor
The EMBO Journal
w.teale@embojournal.org

We realize that it is difficult to revise to a specific deadline. In the interest of protecting the conceptual advance provided by the work, we recommend a revision within 3 months (5th Aug 2025). Please discuss the revision progress ahead of this time with the editor if you require more time to complete the revisions. Use the link below to submit your revision:

Arbitrating Referee:

As I saw, Refs 1 and 3 were quite positive, praising the quality of the study and its importance with relatively minor questions to be addressed. I think the paper was previously rejected because of the strong negative comments from Ref.2.

Ref2 has raised three major points as critiques: first, the specificity of interaction of the tested ligands in picking up their molecular targets. 2nd, the concentrations used in testing the compounds in cells vs. those of the endogenous levels as viewed by Reviewer. 3rd, lack of clarity as to how the affected targets led to the observed changes in cellular responses.

In my view, the Authors have addressed points 1 and 2 with extra new data and very convincing arguments related to the endogenous levels of the sphingosine and its derivatives. I also think that they partially addressed the 3rd point with extra data, although the mechanistic connection between the sphingosine targets and the biological effects have not been all worked out. Still, I believe that the Authors have made a very strong case rebutting Ref2. Their arguments that ongoing studies focus on some of the mechanistic connections to be worked out is quite reasonable.

The Authors also included extra experimental data to satisfy the minor comments of Refs 1 and 3 and I think the revised ms. is suitable for publication in EMBO.

All editorial and formatting issues were resolved by the authors.

Dear Aimee,

I am pleased to inform you that your manuscript has been accepted for publication in the EMBO Journal - congratulations!

Best wishes,

William

William Teale, PhD
Editor
The EMBO Journal
w.teale@embojournal.org
